# The Spatial Spillover Effect and Function Routes of Transport Infrastructure Investment on Economic Growth: Evidence from Panel Data of OECD Members and Partners

**Peiwen Guo [1], Jun Fang [1] and Ke Zhu [1,2,*]**

[1] School of Civil Engineering and Architecture, Wuhan University of Technology, Wuhan 430070, China
[2] Hainan Research Institute, Wuhan University of Technology, Sanya 572000, China
* Correspondence: zhuke2018@whut.edu.cn

**Abstract:** Transport infrastructure connectivity is a spatial basis for economic development and the spillover and feedback effects of transport infrastructure investment (TII) have become an impetus for economic growth (EG). However, existing research does not consider the spatial effects of TII on the gross EG and the multiple effects of TII on EG structures. To explore the spatial relationships and the functional routes between TII and EG, the spatial Durbin model (SDM) was used to empirically analyze the spatial spillover effect of TII on EG from geographical and economic perspectives based on panel data from 2007 to 2019 of 35 members and partners of the Organization for Economic Cooperation and Development (OECD+). On this basis, a structural equation model (SEM) was established to reveal the multiple mediating effects of TII on EG. Results show that (1) the SDM–SEM hybrid method can model the spatial spillover effect and function routes of TII on EG based on theoretical analysis and empirical research; (2) according to empirical analysis of the SDM model, the spatial spillover effect in high-income OECD+ countries shows a positive effect under the economic distance, while that in the upper-middle-income countries has a negative effect; (3) an empirical analysis of the function route model implies that TII in high-income OECD+ countries exerts multiple mediating effects and it mainly affects EG indirectly by means including industrial structure (IS), and the rate of contribution of the key function route FR3 is 67.25%. The following suggestions are proposed: (1) it is necessary to enhance the intensity of effective investment in transport infrastructure, focus on weak links of transport infrastructure, and pay attention to investment in burgeoning fields of the OECD+ countries; (2) differentiated TII strategies are required to be formulated according to development of OECD+ countries with different income levels; (3) it is necessary to give full play to the spatial spillover effect and multiple mediating effects of TII on EG and the TII structure should be optimized, so as to improve the economic benefits of TII.

**Keywords:** transport infrastructure investment; economic growth; the Organization for Economic Cooperation and Development; spatial Durbin model; structural equation model

**MSC:** 62P20; 91D25



## 1. Introduction

The transport infrastructure investment (TII) is significant for economic growth (EG) [1,2]. TII serves as a "catalyst" for economic globalization [3,4], and high-quality transport infrastructure provides efficient and sustainable economic and trade services and tightens the relations of global economy and trade [5,6]. Continuous TII also represents an important impetus of EG in members and partners of the Organization for Economic Cooperation and Development (OECD+) [3,7,8]. Development in OECD+ countries is listed in Table 1, and the gross domestic product (GDP) was 73.47 trillion dollars in 2019, accounting for 86.77% of global GDP, of which TII was 1.20 trillion. The economy and trade have been connected increasingly tightly in OECD+ countries, forming the most dynamic economic aggregates.

In such context, how to promote EG through TII in OECD+ countries has become a system engineering issue and the focus of much research among numerous scholars [9,10].

**Table 1.** TII and GDP of OECD+ countries in 2019 (unit: 100 million dollars).

| Classification | Country [1] | TII [2] | GDP [3] | Country | TII | GDP |
|---|---|---|---|---|---|---|
| Members | Australia | —— | 14,910.25 | South Korea | 224.00 | 16,378.50 |
| | Austria | 26.48 | 4144.24 | Latvia | 2.87 | 307.28 |
| | Belgium | 21.05 | 4947.79 | Lithuania | 5.04 | 481.74 |
| | Canada | 102.43 | 16,961.63 | Luxembourg | 5.70 | 673.14 |
| | Chile | —— | 2620.81 | Mexico | 35.39 | 12,527.32 |
| | Colombia | —— | 3213.94 | Netherlands | —— | 8402.50 |
| | Czechia | 25.26 | 2155.95 | New Zealand | 13.80 | 2021.51 |
| | Denmark | —— | 3346.38 | Norway | 65.72 | 4064.68 |
| | Estonia | 2.91 | 270.79 | Poland | 35.94 | 5701.15 |
| | Finland | 22.85 | 2547.40 | Portugal | —— | 2223.62 |
| | France | 245.00 | 26,168.12 | Slovakia | —— | 990.88 |
| | Germany | 290.71 | 36,003.39 | Slovenia | 5.91 | 502.70 |
| | Greece | 9.84 | 2037.51 | Spain | 65.11 | 13,245.61 |
| | Hungary | 36.44 | 1469.69 | Sweden | 48.23 | 5498.21 |
| | Iceland | 1.05 | 208.47 | Switzerland | 88.93 | 7581.68 |
| | Ireland | 10.45 | 3707.81 | Türkiye | —— | 9974.37 |
| | Israel | —— | 3530.63 | United States of America (USA) | 1216.85 | 199,254.36 |
| | Italy | 99.27 | 19,184.83 | United Kingdom of Great Britain and Northern Ireland (GBR) | 263.74 | 31,914.93 |
| | Japan | 579.20 | 45,690.54 | | | |
| Key partners | Brazil | —— | 18,196.83 | Indonesia | —— | 10,493.30 |
| | China | 8158.90 | 142,963.44 | South Africa | —— | 3587.12 |
| | India | —— | 26,857.48 | —— | —— | —— |
| Observation countries | Bulgaria | 7.50 | 574.44 | Georgia | 5.56 | 177.58 |
| | Croatia | 5.55 | 571.90 | Montenegro | 0.36 | 47.80 |
| | Albania | 2.03 | 129.68 | Russian Federation | 155.32 | 14,615.50 |
| | Azerbaijan | 9.71 | 536.13 | Serbia | 8.87 | 456.16 |
| | Belarus | 16.01 | 590.13 | —— | —— | —— |

[1] Name of the country referred to the *UN Blue Book*. [2] Data of TII results from the transport infrastructure investment and maintenance spending of the International Transport Forum. Source: https://stats.oecd.org/Index.aspx?&datasetcode=ITF_INV-MTN_DATA, accessed on 1 January 2023. [3] Data of GDP results from the World Development Indicators of the World Bank. Source: https://datatopics.worldbank.org/world-development-indicators/, accessed on 1 January 2023.

Baron and Kenny [11] have discussed the effect of TII on EG in the frameworks of the endogenous growth theory and the new economic geography theory, thus, theoretically proving the presence of complex correlation mechanisms between TII and EG [12]. To estimate the effect of TII, the correlations between TII and EG have been discussed through Granger causality tests and cointegration analysis [13]. To overcome drawbacks of direct correlation analysis, TII is separated from influencing factors of EG to establish an econometric model of multiple variables [14]. In terms of research into the data structure of TII, the commonly used methods include cross-section data, time series data, panel data, and spatial panel data (SPD) models. For the research on the spatial spillover effect of TII, the effect on the one hand has significant time delay and, therefore, shows certain differences in short-term and long-term responses [15]; on the other hand, the spatial spillover effect of TII also demonstrates the regional economic gap [16], and the promotional effect of TII on EG is highly heterogeneous in countries with different income levels [12,17]. Previous research has shown that the economic benefit of TII is highly correlated with the spatial location [17] and it is not only dependent on the investment scale but also determined by the level of economic development [11], so, the TII differs greatly in countries with different income levels [18]. Therefore, studying the dynamic spatial economic relationship between TII and EG and their mechanism of action in OECD+ countries with different income levels is of importance when trying to improve the TII quality in OECD+ countries [7].



Although scholars have proved the presence of the spatial spillover effect of TII on EG by selecting different data [19], different models, and different variables, EG is a complex system with a nonlinear structure [17,20]. To study the essential features and intrinsic rules of the complex system of EG [21], it is necessary to determine the relational routes of TII with EG from the perspective of underlying logic [18]. With the integrated development of spatial econometrics, the existing regression models can further characterize the complex structures between variables based on quantifying influences of spatial correlations [22]. Especially in the research paradigm with multiple explained and explanatory variables, the structural equation model (SEM) can be used to calculate the effect of latent variables that cannot be observed directly, which provides a new path for studying the complex relationship and mechanism of action between TII and EG in OECD+ countries [18,23].

The current research proposed a method integrating the spatial Durbin model (SDM) and the SEM to explore the spillover effect and routes whereby TII influences EG. The effectiveness and applicability of the proposed method were verified taking the panel data of 35 OECD+ countries from 2007 to 2019 as examples. The proposed method based on the SDM and SEM enshrines the following innovations: (1) the SDM was used to empirically estimate the spatial spillover effect of TII on EG and study the spatial correlations between TII and EG in countries with different income levels; (2) the SEM was adopted to identify the multiple mediating effects of TII on EG and discuss the modes of action and key functional routes of TII on EG; (3) compared with other research, the method can deal with the heterogeneity of research conclusions due to the multilevel research objects on the basis of studying the spatial spillover effect and functional routes of TII on EG.

The remainder of the paper is organized as follows: Section 2 introduces the SDM, SEM, and SDM–SEM hybrid model; Section 3 discusses the research variables and research objects; the SDM–SEM model is developed in Section 4, in which the detailed analysis steps are also proposed; Section 5 covers empirical analysis of the spatial spillover effect and multiple mediating effects of TII on EG; in Section 6, we draw our conclusions arising from the research; in Section 7, we propose policy suggestions based thereon.

## 2. Research Methods

### 2.1. Spatial Durbin Model

Spatial econometrics aims to solve special problems incurred by space in the statistical analysis model of regional science and it entails variables related to the space, distance, or spatial structure. In terms of the data structure, the spatial panel data model is the main form of the spatial econometric model [24]. Compared with the general panel data model and spatial cross-section data model [21], the spatial panel data model considers not only individual heterogeneity but, also, spatial correlations between individuals [25]. At present, the spatial panel data model has developed from the static panel data model to its dynamic form [25] and empirical research has been conducted into regional economies, regional markets, the labor economy, and the public economy [26]. Regarding model setting, the spatial econometric model includes mainly the SDM, spatial lag model, and spatial error model [26]. The SDM is a combined and expanded form of the spatial lag model and spatial error model. The SDM considers the spatial correlations between explanatory and explained variables from the perspective of different spatial weight matrices [27].

The research discusses the SDM of panel data, which divides all effects of TII into direct and indirect effects. Direct effects are used to discuss the contribution of TII that is regarded as an individual asset to EG. As for indirect effects, they are used to study the externalities of TII, that is, the spatial spillover effect of TII. In summary, SDM performs well in evaluating the spatial effects of variables, while it has poorer explanatory power with regard to the functional routes and mechanisms of influence of variables, so it fails to depict the complex routes between variables, making it necessary to undertake collaborative analysis by combining this with other functional route models and methods.

### 2.2. Mediation Model

Baron and Kenny [28] estimated the direct and indirect effects through linear regression and took the lead to apply the mediation analysis to the social sciences research. A mediation model can be used to reveal the influencing process and mechanism of action of independent variables on dependent variables [29]. Compared with other research into the influence and action of variables, mediation analysis can provide more in-depth results [30]. With the development of the statistical theory and the progress of analytical software, many variants of mediation models have emerged, including the mediation model with categorical variables, multiple mediation model, multilevel mediation model, mediated moderation model, and moderated mediation model [31]. In general, mediation models provide basic conditions for the quantitative research and relation analysis of complex variables [32].

The quantitative research in complex scenarios generally calls for establishment of a multiple mediation model to elucidate the effect of independent variables on dependent variables [33]. The multiple mediation model involves lots of variables and complex routes and, therefore, commonly needs to use other tools to investigate such routes [34]. The current research uses SEM to analyze routes, which is characterized by advantages including simultaneous tests on multiple influencing routes and decomposition and comparison of direct and indirect effects [35], providing a new scenario for the multiple mediation model.

### 2.3. Analysis Framework: The Hybrid Model

By combining advantages of the SDM and SEM, the research developed the SDM–SEM hybrid model for the spatial spillover effect and function routes of TII on EG. Spatial panel data were selected and the SDM was used to construct the spatial econometric model so as to quantify the spatial spillover effect of TII on EG [36]. Considering that the SDM cannot describe complex routes between variables in route analysis, a multiple mediation model based on the SEM was built [37]. On this basis, the indirect, direct, and total effects of variables in complex structures were calculated according to the route coefficient [38,39] to verify the diverse functional routes of TII on EG. Figure 1 illustrates the detailed procedures of the two models.

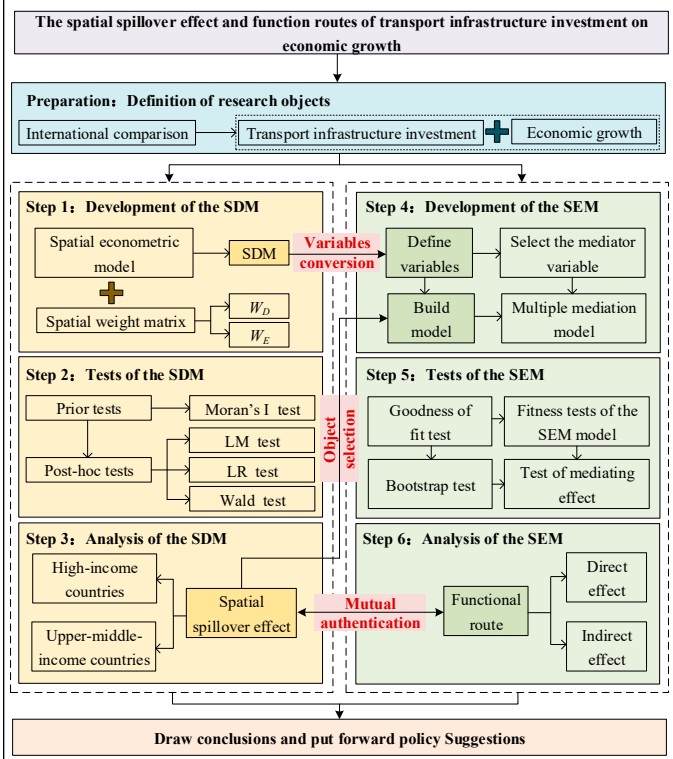

**Figure 1.** Work procedures.

### 3. Research Variables and Objects

*3.1. Research Variables*

3.1.1. Selection of Research Variables

EG is not only affected by TII but also related to the local economic development level [40], industrial structure [41], labor input [42], and level of opening up.

(1).　Selection of the explained variable

The explained variable of the SDM model is the EG. The research focus of the index of EG has gradually shifted from the gross increase to the per capita level. Coase proposed that more attention needs to be paid to the per capita level of EG [18], so the GDP per capita is selected here to measure EG.

(2).　Selection of the core explanatory variable

The core explanatory variable of the SDM is the TII. In previous research, scholars mainly used public investment to characterize TII while ignoring the effect of private capital, which may incur systematic errors [43]. Statistical data of the TII, maintenance spending, and capital value are selected in the research, which cover the total investment in the highways, railways, inland waterways, harbors, and airports [5]. These include all financing sources, maintenance spending funded by public administrative departments, and the capital value of transport infrastructure [10]. These statistical data are used to reduce the systematic errors incurred during data collection.

(3).　Selection of control variables

EG is not only affected by TII but also related to the macroscopic economic development level, IS, labor input, and level of opening up [13]. Considering this, the control variables of the SDM are set in light of the following four perspectives: (i) Level of urbanization measured by urban population (UP); Herrendorf and Valentinyi [44] believe that regions with large UP have a greater density of economic activities, which form a diverse agglomerated economy and attract more capital and talent, thus promoting EG. Therefore, the research uses the proportion of urban population in the total population to represent UP. (ii) Industrial structure (IS); Dong [45] considers IS as an important endogenous variable of EG and IS not only intuitively reflects the coordination degree of each sector in the national economy but also significantly influences EG, so the proportion of the service industry in the GDP is selected to measure IS. (iii) Labor input (LI); the traditional EG theory regards LI as an important factor that drives EG [46]. The research takes the annual labor force participation rate for ages 15 to 24 years across the whole of society in each region as an index to measure LI [42,47]. (iv) Trade (TR); TR has a significant positive effect on EG. The proportion of trade in GDP is selected to measure the TR [48,49].

(4).　Selection of mediators

Transport infrastructure can drive EG via multiple routes consisting of the merchandize export (ME) and consumer expenditure (CE).

CE: The Keynesian school posited the influences of CE on EG into the system of economic theories and pointed out that changes in the CE structure may affect EG [50]. Economists have tested influences of consumption structure upgrading on EG from different perspectives and found that enlarging the TII scale can optimize the CE structure and, thus, influence EG [51].

ME: According to the traditional viewpoints, the contribution of ME to EG is the direct contribution of net exports to GDP. It has been proposed in previous research that improving TII is conducive to enlarging the quantity of ME and optimizing the ME structure, thus, greatly promoting EG [52,53].

3.1.2. Mechanism of Actions of Variables

How does the EG level improve through macroscopic variables including TII from the theoretical perspective? How are the function routes of TII on EG identified? To answer

these questions, it is necessary to establish the whole theoretical framework of TII and EG from peripheral spillover to internal feedback and systematically investigate the trends and characteristics of TII in promoting EG, as shown in Figure 2.

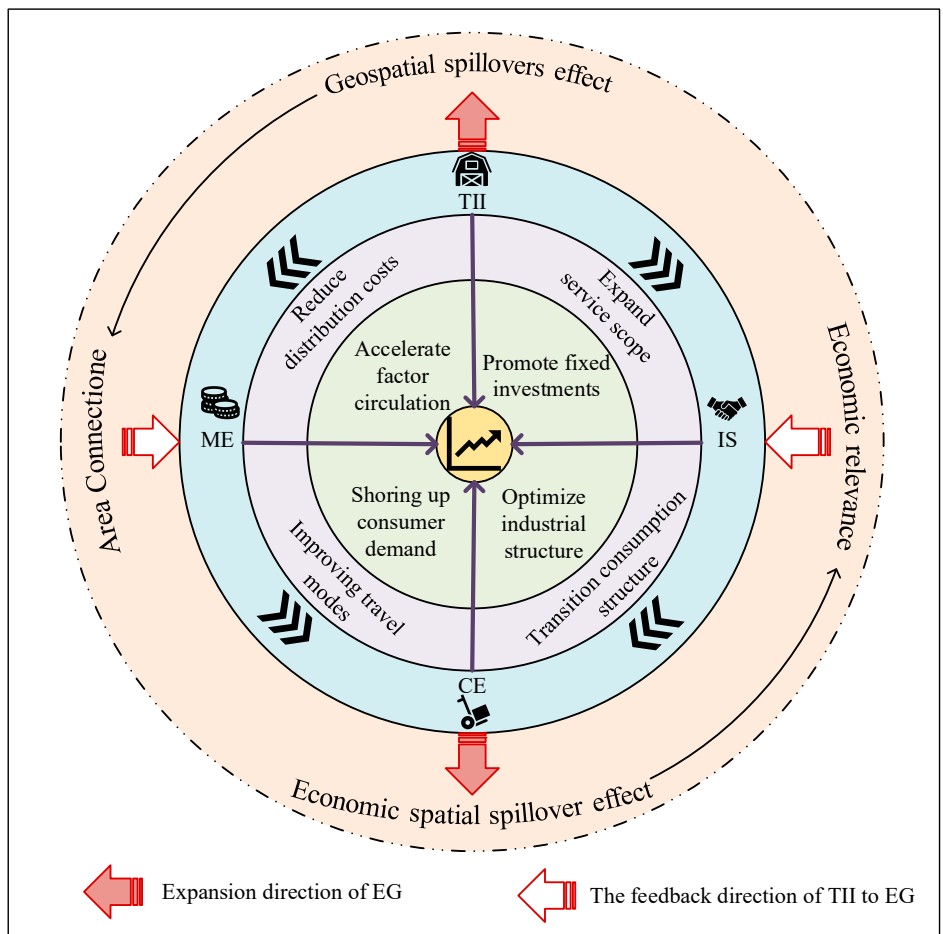

**Figure 2.** Mechanism of actions of TII on EG.

From the interregional perspective, the economic development level, transport infrastructure level, IS, labor input, and opening-up level of different countries show spatial heterogeneity and they function together to produce different economic agglomeration effects [54]. As a result, the spatial spillover effect of TII exhibits different action directions and outcomes.

From the intraregional perspective, TII, as one of the main impetuses for driving EG, does not affect continuous EG via a single route, but strengthens the forward and backward correlations of various sectors based on the accessibility of transport infrastructure. This forms multiple network functional routes [55].

### 3.2. Research Objects

3.2.1. Data Sources

Data from the World Development Indicators (WDI) and International Transport Forum Databases (TFD) from 2007 to 2019 were used in the research; because of data omissions and different statistical quality indicators, 35 OECD+ countries were selected as research objects (Table 2).

**Table 2.** Sources of variables.

| Variable | Meaning | Abbreviation | Data Source |
|---|---|---|---|
| Explained variable | GDP per capita (constant 2015 US$) | EG [1] | WDI |
| Core explanatory variables | Transport infrastructure investment and maintenance spending | TII [2] | International Transport Forum |
| Control variables | Urban population (% of total population) | UP | |
| | Services, value added (% of GDP) | IS | |
| | Labor force participation rate for ages 15 to 24 years, total (%) (modelled ILO estimate) | LI | |
| | Trade (% of GDP) | TR | WDI |
| Mediators | Households and NPISHs final consumption expenditure (current US$) | CE | |
| | Merchandize exports (current US$) | ME | |

[1] Data of GDP results from the World Development Indicators of the World Bank. Source: https://datatopics. worldbank.org/world-development-indicators/, accessed on 1 January 2023. [2] Data of TII results from the transport infrastructure investment and maintenance spending of the International Transport Forum. Source: https://stats.oecd.org/Index.aspx?&datasetcode=ITF_INV-MTN_DATA, accessed on 1 January 2023.

To eliminate heteroscedasticity and reduce influences of fluctuations in variable data, natural logarithms were taken for all nonproportional data. The descriptive statistics of each variable can be found in Table 3. The standard deviation of EG is close to 1, indicating that the economic growth of different countries is significantly different. The standard deviation of TII, CE, and ME is close to 2, which shows that the development of merchandize export and consumer expenditure is not balanced in these countries. Comparatively speaking, there is no obvious difference between the development status of UP, IS, LI, and TR in these countries.

**Table 3.** Descriptive statistics of variables from 2007 to 2019.

| Variable | Number | Mean | Standard Deviation | Min. | Median | Max. |
|---|---|---|---|---|---|---|
| ln *EG* | 455 | 9.983 | 0.902 | 7.971 | 10.14 | 11.63 |
| ln *TII* | 455 | 21.62 | 1.882 | 17.45 | 21.41 | 27.29 |
| ln *UP* | 455 | 4.281 | 0.173 | 3.811 | 4.305 | 4.585 |
| ln *IS* | 455 | 4.093 | 0.175 | 3.074 | 4.111 | 4.383 |
| ln *LI* | 455 | 3.719 | 0.295 | 3.114 | 3.703 | 4.417 |
| ln *TR* | 455 | 4.476 | 0.506 | 3.194 | 4.452 | 5.940 |
| ln *CE* | 455 | 25.812 | 1.905 | 22.740 | 25.755 | 30.240 |
| ln *ME* | 455 | 25.270 | 1.797 | 20.750 | 25.600 | 28.550 |

### 3.2.2. Definition of Research Objects

To determine the evolution characteristics of TII and EG in OECD+ countries with different income levels, OECD+ countries were classified into upper-middle-income countries and high-income countries according to the income level based on GNI per capita in USD according to the World Bank. The logarithm of the GDP per capita on the abscissa represents the economic development level while the logarithm of TTI on the ordinate denotes the investment strength. The TII of high-income countries and upper-middle-income countries is, separately, shown in Figure 3.

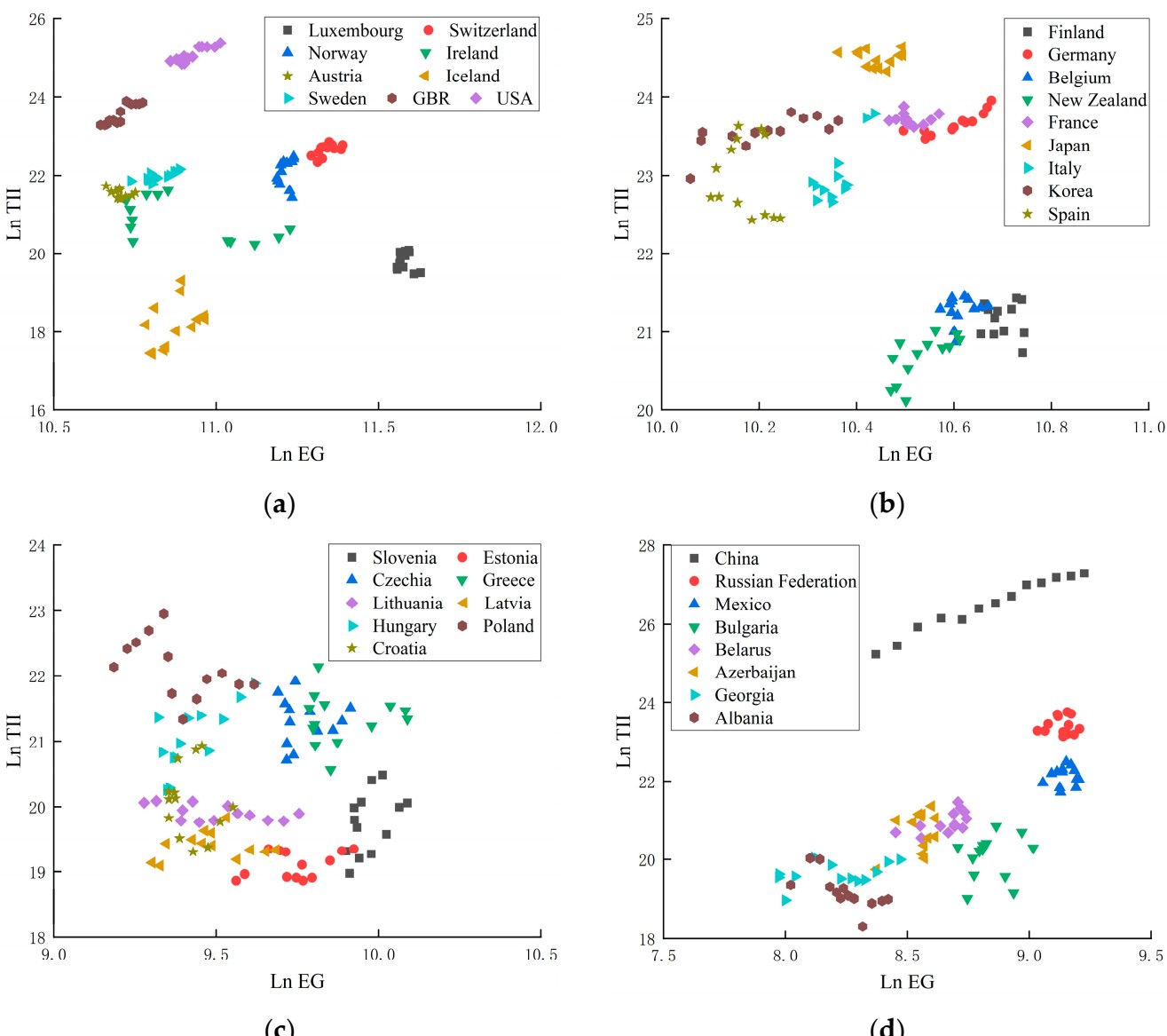

**Figure 3.** TII and EG in OECD+ countries with different income levels from 2007 to 2019. (**a**) The first group of high-income OECD+ countries. (**b**) The second group of high-income OECD+ countries. (**c**) The third group of high-income OECD+ countries. (**d**) The group of upper-middle-income OECD+ countries.

Figure 3 indicates that TII in high-income OECD+ countries increases with EG. Taking the United States as an example, its gross TII scale is the largest and TII is highly correlated with EG in terms of the trend. TII in South Korea and Germany also grows steadily with EG and their TII levels are ranked top among OECD+ countries.

As illustrated in Figure 3, TII changes in diverse manners with EG in upper-middle-income OECD+ countries. For example, China takes the leading position in TII, which is significantly positively correlated with EG. However, TII is highly decoupled with EG in Mexico and shows a decreasing trend.

In summary, the effect of TII on EG in OECD+ countries can be affected by the economic development level. In high-income OECD+ countries, TII changes to a greater extent with EG; while, changes in TII show multiple trends with EG in upper-middle-income OECD+ countries. To formulate targeted development strategies for countries with

different economic development levels, OECD+ countries were classified into two types of research objects: high-income and upper-middle-income ones.

## 4. Model Development

### 4.1. Step 1: Development of the Spatial Econometric Model

To identify the correlations between TII and EG, the SDM was used to quantify the spatial spillover effect of TII on EG. The initial conceptual model is expressed as Equation (1).

$$
\begin{aligned}
\ln EG_{it} = {} & \rho W \ln EG_{it-1} + \beta_1 \ln TII_{it} + \beta \ln X_{it} \\
& + \delta_1 W \ln TII_{it} + \delta W \ln X_{it} + u_i + v_t + \varepsilon_{it}
\end{aligned}
\tag{1}
$$

where subscripts $i$ and $t$, separately, represent dimensions of cross section and time ($i = 1, 2, \cdots, N$ and $t = 1, 2, \cdots, T$); $\ln EG_{it}$ denotes the logarithmic value of the explained variable (EG); $\ln TII_{it}$ is the logarithmic value of the core explanatory variable (TII); $X_{it}$ represents the control variable; $W$ is the spatial weight matrix; $\rho$, $\beta$, and $\delta$ are all model parameters to be estimated; $u_i$ represents the effect of spatial characteristics of spatial units; $v_t$ refers to the temporal trend effect of research objects; and $\varepsilon_{it}$ is an error term.

Specifically, $W$ can be estimated by geographical distance and economic distance. The spatial effect is attenuated with the increasing geographical distance [56], so the reciprocal of the geographical distance is generally adopted to establish the inverse distance spatial weight matrix $W^D$ [57], as shown in Equation (2). Considering the closeness of the size of the economy between different regions is not significantly correlated with the geographical location, the inverse distance spatial weight matrix $W^E$ of economy is established by taking the reciprocal of the difference in GDP per capita in various regions as the weight, as shown in Equation (3).

$$
w_{ij}{}^D = \begin{cases} 1/d_{ij} & i \neq j \\ 0 & i = j \end{cases}
\tag{2}
$$

where $d_{ij}$ denotes the regional distance calculated based on the longitude and latitude of the capital.

$$
w_{ij}{}^E = \begin{cases} 1/|g_i - g_j| & i \neq j \\ 0 & i = j \end{cases}
\tag{3}
$$

where $g_i$ and $g_j$ represent the GDP per capita in regions $i$ and $j$.

### 4.2. Step 2: Tests of the Spatial Econometric Model

To guarantee accuracy of the spatial effect and the settings of the spatial model, it is necessary to perform prior tests and post hoc tests on the spatial econometric model.

#### 4.2.1. Prior Tests on Spatial Correlations

Prior tests on spatial correlations are to test whether there are statistically significant spatial effects before setting the spatial econometric model and, generally, include Moran's I tests [58], joint LM tests [59], maximum likelihood LM-lag tests, maximum likelihood LM-error tests [60], robust LM-lag tests, and robust LM-error tests [61]. Among them, Moran's I tests and the improved tests are applied most widely and exhibit the optimal effects. Therefore, Moran's I and local Moran's I were adopted to perform model tests on spatial data distribution.

(1). Moran's I: a positive Moran's I indicates spatial agglomeration, while a negative value suggests spatial decentralization. The larger the absolute value, the stronger the spatial correlations, as shown in Equations (4) and (5).

$$\text{Moran's I} = \frac{N}{S_0} \frac{\sum\limits_{i=1}^{N} \sum\limits_{j=1}^{N} w_{ij}(y_i - \overline{y})(y_j - \overline{y})}{\sum\limits_{j=1}^{N} (y_i - \overline{y})^2} \tag{4}$$

$$S_0 = \sum_{i=1}^{N} \sum_{j=1}^{N} w_{ij} \tag{5}$$

where $N$ and $S_0$, separately, denote the number of spatial units and the sum of all elements in the spatial weight matrix; $y_i$ is the observed value of variable y in a spatial unit $i$; $\overline{y}$ is the mean value of variable y; and $w_{ij}$ represents the element in the spatial weight matrix.

(2).  Local Moran's I: the local Moran's I statistic is proposed by improving Moran's I and it is used to calculate the contributions of observed values of each spatial unit, as expressed in Equations (6) and (7).

$$\text{Local Moran's I}_i = \frac{y_i - \overline{y}}{S_i^2} \sum_{j=1, j \neq i}^{N} w_{ij}(y_j - \overline{y}) \tag{6}$$

$$S_i^2 = \frac{\sum\limits_{j=1, j \neq i}^{N} w_{ij}}{N - 1} - \overline{y}^2 \tag{7}$$

### 4.2.2. Post Hoc Tests on Spatial Correlations

Post hoc tests are generally conducted to test estimation results of the econometric model, to determine whether the model has effectively quantified the spatial effect. Post hoc tests are divided into three steps: at first, the Sargan–Hansen statistic is used to test the applicability of the fixed and random effects of the model [62]; then, LM tests are performed to judge whether there is a spatial lag variable or error variable in the model [63]; finally, LR tests and Wald tests are applied to judge whether the spatial panel data model will be degraded from the SDM to the spatial lag model or the spatial error model [64]. Hypotheses associated with post hoc tests are listed in Table 4.

**Table 4.** Hypotheses associated with post hoc tests.

|  | **Test Statistics** | **Alternative Hypothesis** |
|---|---|---|
|  | Sargan-Hansen statistic | Difference in coefficients not systematic |
| LM test | Spatial error<br>Spatial lag | LM test no spatial error<br>LM test no spatial lag |
| LR test | LR test for spatial lag model<br>LR test for spatial error model | Spatial lag model nested in SDM<br>Spatial error model nested in SDM |
| Wald test | Wald test for spatial lag model<br>Wald test for spatial error model | Spatial lag model nested in SDM<br>Spatial error model nested in SDM |

### 4.3. Step 3: Analysis of the Spatial Econometric Model

The spatial econometric model is studied mainly following four steps: firstly, the SDM is preliminarily regressed to study influences of TII and the spatial lag on EG; secondly, the partial differential equation is utilized to decompose the total effect into the direct and indirect effects to estimate the spatial spillover effect of TII on EG; thirdly, robustness tests are conducted to explore the robustness of the evaluation method and the index explanatory power; finally, the characteristics of the spatial spillover effect in OECD+ countries with different income levels are summarized.

### 4.3.1. Regression Analysis of the SDM

To quantify influences of TII on EG in OECD+ countries with different income levels, the maximum likelihood estimation was used to regress the SDM using $W^D$ and $W^E$ [65]. By doing so, the parameter estimation results of variables of countries with different income levels were obtained from perspectives of geographical distance and economic distance. The sensitivity of spatial effects of TII to the geographical distance and economic distance was estimated according to the significance and value of the regression coefficients.

### 4.3.2. Estimation of the Spatial Spillover Effect

The direct effect of the explanatory variable represents the average influence of TII on local EG, while the indirect effect indicates the average influence of TII in neighboring regions on the local EG. A significant indirect effect implies the presence of a spatial spillover effect between regions. The regression coefficient of the SDM fails to directly reflect the spatial effect of the explanatory variable on the explained variable. To solve this problem, the partial differential equation is used to decompose the total effect of the explanatory variable into direct and indirect effects [27].

### 4.3.3. Robustness Tests

Robustness tests were conducted to evaluate the robustness of the evaluation method and the explanatory power of each index [66] that is, whether the evaluation method and the indices retain their consistent and stable explanation of evaluation results when changing certain parameters. The commonly used robustness test methods include variable substitution, changing sample size, subsample regression, adjusting sample periods, and supplementary variable technique.

### 4.3.4. Analysis of the Spatial Spillover Effect

Characteristics of the spatial spillover effect of TII in countries with different income levels were summarized according to regression estimation results of the SDM and estimation results of the spatial spillover effect [67]. Then, the spatial economic ties of TII were studied from perspectives of geographical distance and economic distance.

### 4.4. Step 4: Development of the Functional Route Model

The SDM verifies the presence of the spatial spillover effect of TII in dimensions of geographical distance and economic distance, while failing to quantify the complex route relationships between TII and EG. Therefore, a multiple mediation model of TII on EG is established based on the SEM after evaluating the SDM. The conceptual model is illustrated in Figure 4.

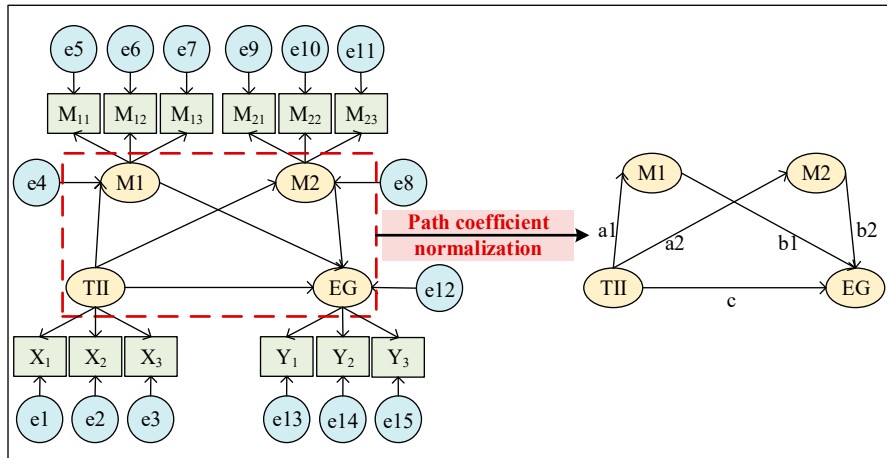

**Figure 4.** The conceptual model of mediating effects.

According to Figure 4, routes through which TII influences EG are called direct routes, while those by which TII acts on EG via mediators M1 and M2 are termed indirect routes.

### 4.5. Step 5: Tests of the Function Route Model

The overall fitness of the model was tested to ensure that the presumptive model matches the empirical data. The mediating effects of the routes were investigated to identify the key function route.

#### 4.5.1. Fitness Tests of the Multiple Mediation Model

Fitness tests aim to ensure the explicability of the research conclusions. Fitness indices are also known as fit indices, which are used to evaluate the matching degree between data and the presumptive model. The test standards [68] are summarized in Table 5.

**Table 5.** Fitness test standards of the SEM.

| Tests Statistic | | Meaning | Fitness Test Standard |
|---|---|---|---|
| Absolute fit indices | CMIN/DF | Discrepancy divided by degree of freedom | <3.00 |
| | P | Probability | >0.05 |
| | RMR | Root mean square residua | <0.05 |
| | RMSEA | Root mean square error of approximation | <0.05 |
| | GFI | Goodness of fit index | >0.90 |
| | AGFI | Adjusted goodness of fit index | >0.90 |
| Relative fit indices | IFI | Incremental fit index | >0.90 |
| | CFI | Comparative fit index | >0.90 |
| | NFI | Normal fit index | >0.90 |
| | TLI | Tucker-Lewis index | >0.90 |

If the test statistics do not conform to the fitness standards, modification indices need to be used to modify and optimize the model, to reach the standards of indices for fitness evaluation.

#### 4.5.2. Bootstrap Tests on Functional Routes

Statistical tests on the mediating effects mainly include the causal method [69,70], difference in coefficients, product of coefficients [71], and bootstrap method [72]. Among them, the bootstrap method is not limited by the sampling distribution patterns and is applicable to medium and small sample sizes and all kinds of mediation models with a wide applicable range [72]. Therefore, the research uses Amos software to estimate the coefficient of functional routes and applies the bootstrap method to assess the mediating effects of variables and measure the direct, indirect, and total effects to quantify the mediation model of TII on EG.

### 4.6. Step 6: Analysis of the Function Route Model

According to the mediation results estimated in Step 5, functional routes of TII on EG were studied from the direct and indirect routes to judge the key functional route based on the rate of contribution of routes. Moreover, the planning directions of TII were proposed and combined with the analysis results of the SDM to maximize the economic benefits of TII.

## 5. Empirical Analysis Results

### 5.1. Step 1: Development of the Spatial Econometric Model

Variance inflation factor (VIF) is one of the important methods to measure multicollinearity in multiple linear regression models, which is very suitable for economic and social multicollinearity analysis. VIF of TII, UP, IS, LI, and TR is calculated as 1.81, 1.76, 1.46, 1.44, and 1.09; mean VIF is 1.51 and after VIF screening (VIF > 10), all variables are retained.

In the present research, the explained variable is EG; the core explanatory variable is TII; the control variables include UP, IS, LI, and TR. To estimate the sensitivity of the spatial spillover effect, the SDM was built based on $W^D$ and $W^E$ as follow:

$$
\ln EG_{it} = \rho W^D \ln EG_{it-1} + \beta_1 \ln TII_{it} + \beta_2 \ln UP_{it} + \beta_3 \ln IS_{it} + \beta_4 \ln LI_{it}
$$
$$
+ \beta_5 \ln TR_{it} + \delta_1 W^D \ln TII_{it} + \delta_2 W^D \ln UP_{it} + \delta_3 W^D \ln IS_{it} \tag{8}
$$
$$
+ \delta_4 W^D \ln LI_{it} + \delta_5 W^D \ln TR_{it} + u_i + v_t + \varepsilon_{it}
$$

$$
\ln EG_{it} = \rho W^E \ln EG_{it-1} + \beta_1 \ln TII_{it} + \beta_2 \ln UP_{it} + \beta_3 \ln IS_{it} + \beta_4 \ln LI_{it}
$$
$$
+ \beta_5 \ln TR_{it} + \delta_1 W^E \ln TII_{it} + \delta_2 W^E \ln UP_{it} + \delta_3 W^E \ln IS_{it} \tag{9}
$$
$$
+ \delta_4 W^E \ln LI_{it} + \delta_5 W_E \ln TR_{it} + u_i + v_t + \varepsilon_{it}
$$

where $EG_{it}$, $TII_{it}$, $UP_{it}$, $IS_{it}$, $LI_{it}$, and $TR_{it}$, separately, represent the economic development level, TII, level of urbanization, industrial structure, scale of labor population, and foreign trade. $W^D$ and $W^E$, separately, denote the inverse distance spatial weight matrices of geology and economy; $\rho$, $\beta$, and $\delta$, separately, refer to the model parameters to be estimated; $u_i$ and $v_t$, separately, represent the effect of spatial characteristics of spatial units and the temporal trend effect of research objects; and $\varepsilon_{it}$ is the error term.

### 5.2. Step 2: Tests on the Spatial Econometric Model
#### 5.2.1. Prior Tests on Spatial Correlations

According to Equations (4) and (5), the Moran's I of TII and EG in $W^E$ in each OECD+ country from 2007 to 2019 was calculated, and the results are shown in Figure 5.

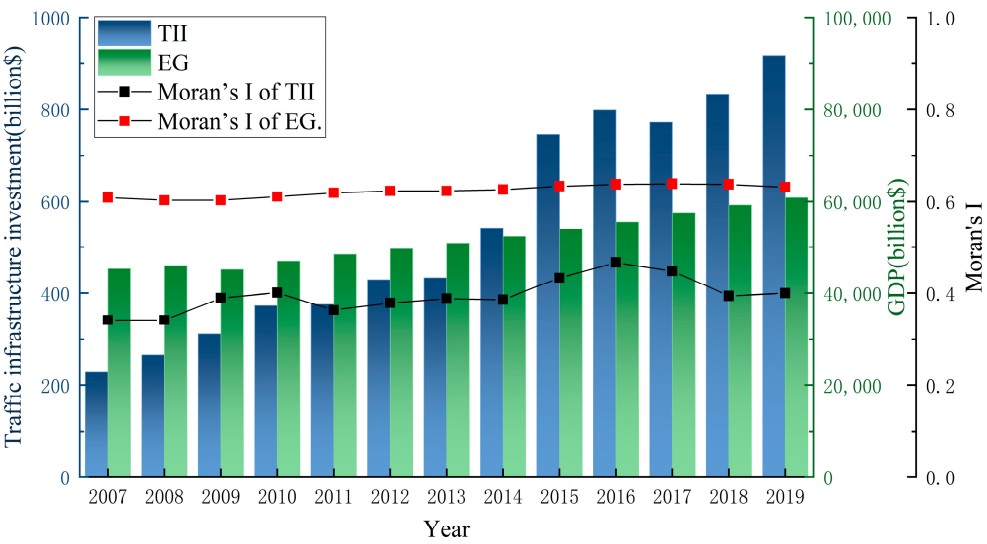

**Figure 5.** Aggregate trends and Moran's I of TII and EG from 2007 to 2019.

As shown in Figure 5, Moran's I is always positive, which indicates that TII and EG have a positive spatial agglomeration effect. Moran's I of TII increases in fluctuation, suggesting that the degree of spatial agglomeration of variables is enhanced.

According to Equations (6) and (7), the statistics of local Moran's I could be used as the regression coefficient of scatter plots, in which the abscissa and ordinate, separately, represent variables and the spatial lag vector of variables, as shown in Figure 6.

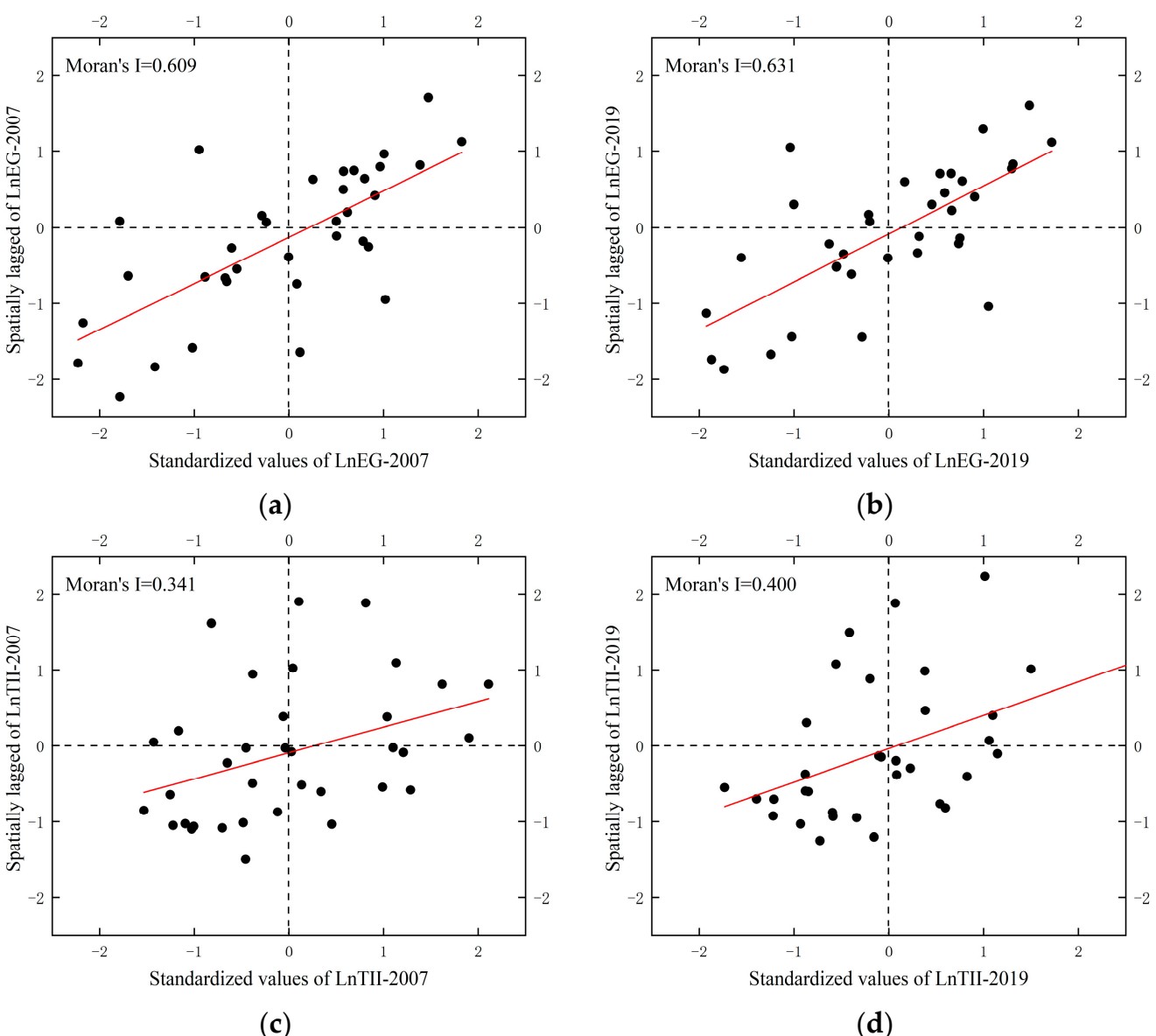

**Figure 6.** Local Moran's I scatter plot of EG and TII in 2007 and 2019. (**a**) Local Moran's I of ln *EG* in 2007. (**b**) Local Moran's I of ln *EG* in 2019. (**c**) Local Moran's I of ln *TII* in 2007. (**d**) Local Moran's I of ln *TII* in 2019.

According to Figure 6, the local Moran's I of TII and EG is mainly distributed in the first and third quadrants, with only a few observed values distributed in the second and fourth quadrants. This indicates that spatial units with high observed values are surrounded by those also having high observed values over the chosen 13-year period.

In summary, TII and EG have a large spatial interaction in each OECD+ country from 2007 to 2019 and show positive spatial correlations.

### 5.2.2. Post Hoc Tests on Spatial Correlations

To select the econometric model of TII and EG, the spatial panel data model was tested according to Table 4 and the test results are summarized in Table 6.

**Table 6.** Diagnostic tests on the spatial panel data model.

| | Test Statistics | Test Results |
|---|---|---|
| | Sargan–Hansen statistic | 22.951 *** |
| Spatial error | Moran's I | 6.015 *** |
| | Lagrange multiplier | 34.552 *** |
| | Robust Lagrange multiplier | 15.461 *** |
| Spatial lag | Lagrange multiplier | 19.160 *** |
| | Robust Lagrange multiplier | 0.069 |
| LR test | LR test for spatial lag model | 70.14 *** |
| | LR test for spatial error model | 110.45 *** |
| Wald test | Wald test for spatial lag model | 202.51 *** |
| | Wald test for spatial error model | 229.30 *** |

Notes: *** Significant at 1% level.

Table 6 shows that the Sargan–Hansen statistic significantly negates the null hypothesis of random effects, implying that the fixed-effect model should be used. LM-Lag, LM-error, and robust LM-error statistics all significantly reject hypotheses of the nonspatial effect, which means that it is reasonable to use the spatial panel data model to determine the spatial effect of TII on EG. The results of LR and Wald tests are significantly positive at the 1% level, separately, rejecting the null hypotheses of "the SDM may be degraded into the spatial error model" and "the SDM may be degraded into the spatial lag model", indicating that it is appropriate to select the SDM. Comprehensive analysis of the diagnostic test results of the spatial panel data model suggests that the fixed-effect SDM model should be used here.

*5.3. Step 3: Analysis of the Spatial Econometric Model*

5.3.1. Regression Estimation of the SDM

Analysis of Steps 1 and 2 shows that the fixed-effect SDM model was selected as the benchmark analysis model in the research. OECD+ countries were classified into two types of research objects: high-income and upper-middle-income ones. Based on $W^D$ and $W^E$, maximum likelihood estimation was adopted for regression estimation of the SDM of the two types of research objects, separately, using Equations (8) and (9). Weighted least squares (WLS) and robust standard errors are sometimes presented as alternative approaches for obtaining reliable standard errors of estimates of regression coefficients in the presence of heteroscedasticity. The use of weighted least squares with robust standard errors is displayed in Table 7.

Analysis of Table 7 implies that TII in high-income countries is underestimated by 21.30%, $(0.108–0.085)/0.108 = 21.30\%$, while that in upper-middle-income countries is underestimated by 53.85%, $(0.104–0.048)/0.104 = 53.85\%$, when only considering the geographical distance compared with coefficient estimation of $W^E$. This finding indicates that ignoring the economic development level may underestimate the contribution of TII to EG. The $W^D$ of the spatial lag of TII is significantly positive, which suggests that geographical distance is a leading factor that affects the spatial spillover effect of transport infrastructure.

It is worth noting that TII and the estimated coefficients of the spatial lag of upper-middle-income countries are both lower than those of high-income countries. This indicates that TII is more active in high-income countries and it is easier to drive EG in other countries. Furthermore, the level of urbanization and industrial structure both significantly stimulate EG and have a positive spatial spillover effect.

**Table 7.** Regression results of the SDM for OECD+ countries with different income levels.

| Variable | High-Income | | Upper-Middle-Income | |
|---|---|---|---|---|
| | $W_H{}^E$ | $W_H{}^D$ | $W_{UM}{}^E$ | $W_{UM}{}^D$ |
| ln *TII* | 0.108 ** [1] | 0.085 ** | 0.104 ** | 0.048 ** |
| ln *UP* | 0.522 | 0.571 * | 1.267 * | 1.072 * |
| ln *IS* | 1.223 | 2.383 ** | 0.168 ** | 0.565 ** |
| ln LI | 0.713 *** | 1.270 *** | −0.008 | 0.082 * |
| ln *TR* | 0.367 ** | 0.291 ** | −0.029 | −0.237 ** |
| L.*W* ln *EG* | 0.578 ** | −6.447 *** | −0.422 ** | −1.663 *** |
| *W*\*ln *TII* [2] | 0.033 | 1.380 *** | −0.004 | 0.068 *** |
| *W*\*ln *UP* | −0.485 | 3.328 | 0.236 | 1.917 |
| *W*\* ln *IS* | −0.687 | 24.878 *** | 0.278 ** | 1.178 *** |
| *W*\*ln LI | 0.427 | 7.039 *** | 0.040 | 0.377 *** |
| *W*\*ln *TR* | 0.037 | 3.615 *** | 0.070 | −0.450 *** |
| Spatial rho [3] | 0.033 *** | 0.076 *** | 0.013 *** | 0.038 *** |
| Variance sigma2_e | 0.086 *** | 0.092 *** | 0.002 *** | 0.001 *** |
| N | 324 | 324 | 96 | 96 |
| r2 | 0.789 | 0.704 | 0.963 | 0.884 |

[1] Standard errors in parentheses: * $p < 0.1$, ** $p < 0.05$, and *** $p < 0.01$. [2] *W*\* refers to the spatial spillover coefficient of variables on other regions. [3] Spatial rho is the spatial spillover coefficient of explained variables on surrounding regions.

### 5.3.2. Estimation of the Spatial Spillover Effect

To estimate the spatial spillover effect, the partial differential equation is adopted to decompose the spatial effect of the SDM. Based on $W^D$ and $W^E$, the total effect of TII on EG is decomposed into direct and indirect effects. The decomposition results are listed in Table 8.

**Table 8.** Estimation results of direct, indirect, and total effects.

| Matrix | Category | Variable | Short-Term Response | | | Long-Term Response | | |
|---|---|---|---|---|---|---|---|---|
| | | | Direct | Indirect | Total | Direct | Indirect | Total |
| $W^E$ | High-income | ln TII | 0.105 ** | 0.027 | 0.132 * | 0.129 *** | 0.17 | 0.299 * |
| | | ln UP | 0.561 | −0.471 | 0.009 | 0.511 | −0.315 | 0.196 |
| | | ln IS | 1.167 | −0.724 | 0.444 | 1.142 | −0.123 | 1.019 |
| | | ln LI | 0.699 *** | 0.391 | 1.090 *** | 0.919 *** | 1.555 *** | 2.474 *** |
| | | ln TR | 0.371 ** | 0.022 | 0.393 *** | 0.434 *** | 0.455 | 0.889 |
| | Upper-middle-income | ln TII | 0.102 ** | −0.006 | 0.096 * | 0.119 ** | −0.067 | 0.052 * |
| | | ln UP | 1.306 | 0.241 | 1.547 | 1.446 | −0.308 | 1.137 |
| | | ln IS | 0.099 | 0.253 | 0.353 | 0.006 | 0.066 | 0.072 |
| | | ln LI | −0.019 ** | 0.038 | 0.019 ** | −0.05 *** | −0.053 *** | −0.104 *** |
| | | ln TR | −0.028 * | 0.075 | 0.046 *** | −0.065 ** | 0.066 | 0.003 |
| $W^D$ | High-income | ln TII | 0.077 ** | 1.275 *** | 1.352 *** | 0.275 *** | −0.081 ** | 0.193 *** |
| | | ln UP | 0.575 ** | 3.049 | 3.624 * | 0.456 | 0.062 | 0.519 |
| | | ln IS | 2.237 ** | 23.091 *** | 25.328 *** | 4.575 *** | −0.955 | 3.620 *** |
| | | ln LI | 1.246 *** | 6.413 *** | 7.659 *** | 0.948 ** | 0.147 | 1.095 *** |
| | | ln TR | 0.279 * | 3.304 *** | 3.583 *** | 0.672 ** | −0.159 | 0.513 *** |
| | Upper-middle-income | ln TII | 0.045 ** | 0.012 *** | 0.057 *** | 0.362 *** | −0.32 ** | 0.042 *** |
| | | ln UP | 1.077 ** | 1.76 | 2.837 * | 1.752 | −0.643 | 1.11 |
| | | ln IS | 0.505 ** | 0.077 *** | 0.582 *** | 3.517 *** | −2.881 | 0.636 *** |
| | | ln LI | 0.08 *** | 0.033 *** | 0.111 *** | −0.814 ** | 0.981 | 0.168 *** |
| | | ln TR | −0.24 | −0.633 ** | −0.874 *** | 0.554 ** | −0.816 | −0.261 *** |

Notes: *** Significant at 1% level; ** significant at 5% level; * significant at 10% level.

Analysis of Tables 7 and 8 shows that the feedback effect of TII in high-income countries is larger under $W^E$; the short-term feedback effect of TII is almost zero while the long-term feedback effect is larger under $W^D$ in upper-middle-income countries. The result suggests that the level of economic development can magnify the promotional effect of TII on EG in

high-income countries to a greater extent, while geographical distance is the main factor that enables TII to promote EG in upper-middle-income countries.

By analyzing Table 8, it shows that when taking high-income countries as an example, the gross economic output in the local area increases by 0.077% in the short term under $W^D$ and the value gradually increases to 0.275% per 1% increase in TII while keeping other conditions unchanged. In addition, the gross economic output in surrounding areas grows by 1.275%, which gradually reduces and then stabilizes at −0.081%, suggesting the simultaneous reduction in the total effect. Under $W^E$, the gross economic output in the local area may increase by 0.105% in the short term and then grow to 0.129%; the gross economic output in surrounding areas increases by 0.027% and then constantly increases to 0.17%, indicating a simultaneous increase in the total effect. Furthermore, the total response of urbanization, industrial structure, labor input, and trade significantly increases in high-income countries under $W^E$.

### 5.3.3. Robustness Tests

To diminish the deviation arising from specific data on the model estimation, the method of shortening the time window was used to exclude statistical data in 2007. Then, panel data from 35 OECD+ countries (2008 to 2019) were used for regression of the spatial Durbin panel model, as shown in Table 9.

**Table 9.** The estimated results of robustness test based on shortening the time window.

| Variable | High-Income | | Upper-Middle-Income | |
|---|---|---|---|---|
| | $W_H{}^E$ | $W_H{}^D$ | $W_{UM}{}^E$ | $W_H{}^E$ |
| ln *TII* | 0.111 *** | 0.088 *** | 0.100 *** | 0.046 *** |
| ln *UP* | 0.468 *** | 0.485 *** | 1.309 *** | 0.943 *** |
| ln *IS* | 1.279 *** | 2.466 *** | 0.071 | 0.717 *** |
| ln *LI* | 0.719 *** | 1.299 *** | 0.000 | 0.147 *** |
| ln *TR* | 0.355 *** | 0.276 *** | −0.031 | −0.190 *** |
| L.*W* ln *EG* | 0.561 *** | −6.665 *** | −0.352 *** | −2.068 *** |
| $W \times$ ln *TII* | 0.042 * | 1.498 *** | −0.023 * | 0.099 *** |
| $W \times$ ln *UP* | −0.576 *** | 2.784 *** | 0.278 * | 2.837 *** |
| $W \times$ ln *IS* | −0.555 | 25.780 *** | 0.186 ** | 1.255 *** |
| $W \times$ ln *LI* | 0.488 *** | 7.329 *** | 0.064 * | 0.462 *** |
| $W \times$ ln *TR* | 0.051 | 3.697 *** | 0.061 | −0.486 *** |
| Spatial rho | 0.033 | 0.083 | 0.028 | 0.035 |
| Variance sigma2_e | 0.085 *** | 0.090 *** | 0.002 *** | 0.001 *** |
| N | 297 | 297 | 88 | 88 |
| r2 | 0.793 | 0.711 | 0.966 | 0.861 |

Notes: *** Significant at 1% level; ** significant at 5% level; * significant at 10% level.

Comparative analysis of Tables 7 and 9 indicates that there is no significant difference in signs and significance of coefficients, so it is inferred that the results of the SDM for the effect of TII on EG are relatively robust.

### 5.3.4. Analysis of the Spatial Spillover Effect

On the one hand, the network effect of transport infrastructure is conducive to promoting spatial agglomeration and decentralization of economic activities and enhancing the geographical contacts and economic ties between regions in OECD+ countries. On the other hand, in OECD+ countries with similar geographical distances and levels of economic development, whether the transport networks are developed or not directly influences the strength of spatial economic ties between them, thus, affecting the level of economic cooperation and development.

(1). Characteristics of the spatial spillover effect in high-income countries

In high-income OECD+ countries, the economic development level magnifies the spatial spillover effect of TII on EG. The spatial networking and density increase in transport infrastructure incur a space–time compression effect, which enhances the frequency and breadth of connections between high-income OECD+ countries and promotes coordinated development in neighboring regions. As a result, the zone of influence gradually extends beyond traditional geographical boundaries and coalesces, thus, forming a benign cycle between TII and EG.

(2). Characteristics of the spatial spillover effect in upper-middle-income countries

In upper-middle-income OECD+ countries, geographical distance enhances the negative spatial spillover effect of TII on EG. That is, increasing TII in a country is conducive to improving EG in the country and its competitiveness, while this intensifies competition between upper-middle-income countries, thus, occupying economic resources of other upper-middle-income OECD+ countries. Meanwhile, the spatial spillover effect of TII in upper-middle-income OECD+ countries attenuates with distance according to the new economic geography theory. As the distance constantly increases, the number of regions with spatial relations reduces continuously and the spatial spillover effect of TII gradually decreases and almost disappears. It is worth noting that in Bulgaria, Albania, Azerbaijan, Belarus, Georgia, Montenegro, and the Russian Federation, as OECD+ "economies in transition" due to their specific socioeconomic characteristics and the closeness of their connections with OECD countries, the spatial spillover effect of transport infrastructure investment is affected to different degrees.

*5.4. Step 4: Development of the Functional Route Model*

Results of the SDM show that the TII exhibits more significant economic benefits in high-income OECD+ countries. Therefore, to further explore functional routes of economic benefits, only high-income OECD+ countries with significant economic benefits are selected as research objects below. As shown in Table 3, the development of ME and CE is not balanced in these countries. In addition, in Table 7, the value of IS is the largest and much larger than 1. The explained and explanatory variables in the SDM are, separately, transformed into dependent and independent variables in the SEM, with ME, IS, and CE as mediators. Moreover, the service industry is taken to represent the direction of development and the transformation of the economy. Driven by high-quality development of the service industry, the service supply is constantly released, which boosts the upgrading of CE. Considering this, the hypotheses in Table 10 are proposed.

**Table 10.** Hypotheses in the mediation model.

| Hypothesis | Explanation |
|:---:|:---:|
| H1 | TII plays a positive effect on EG. |
| H2 | ME plays a positive effect in the relationship between TII and EG. |
| H2a | TII plays a positive effect on ME. |
| H2b | ME plays a positive effect on EG. |
| H3 | IS plays a positive effect in the relationship between TII and EG. |
| H3a | TII plays a positive effect on IS. |
| H3b | IS plays a positive effect on EG. |
| H4 | CE plays a positive effect in the relationship between TII and EG. |
| H4a | TII plays a positive effect on CE. |
| H4b | CE plays a positive effect on EG. |
| H3-H4 | IS plays a positive effect on CE. |

The functional route model of TII on EG is built, as shown in Figure 7.

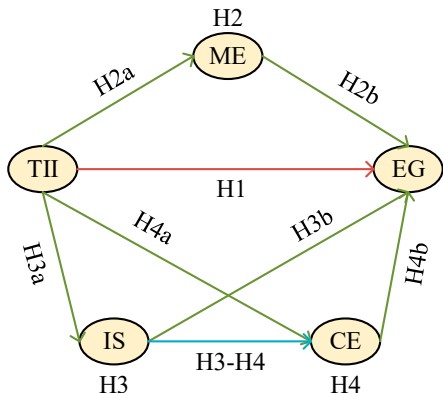

**Figure 7.** The conceptual model of mediating effects of TII on EG.

Various function routes in each subsection in the multiple mediation model are connected to form a direct functional route and four indirect functional routes of TII to EG (Table 11).

**Table 11.** Functional routes of the effects of TII on EG.

| (1) | (2) | (3) | (4) | (5) |
|---|---|---|---|---|
| **Abbreviation** | **Functional Route** | **Direct Effect** | **Indirect Effect** | **Total Effect** |
| FR 1 | TII → EG | √ | | |
| FR 2 | TII → ME → EG | | √ | |
| FR 3 | TII → IS → EG | | √ | (3) + (4) |
| FR 4 | TII → CE → EG | | √ | |
| FR 5 | TII → SI → CE → EG | | √ | |

*5.5. Step 5: Tests of the Function Route Model*

5.5.1. Fitness Tests of the Multiple Mediation Model

Amos software was used to conduct fitness tests and estimate coefficients of functional routes, calculate the overall fitness of the multiple mediation model, and test the degree of matching between the presumptive model and the actual data. The initial model was modified and optimized according to the MI parameters and the final fitting results are listed in Table 12.

**Table 12.** Fitness results of the SEM.

| **Test Statistics** | | **Fitness Test Standard** | **Fitness Test Results** | **Fitness Test Conclusion** |
|---|---|---|---|---|
| Absolute fit indices | CMIN/DF | <3 | 0.008 | Pass |
| | P | >0.05 | 0.928 | Pass |
| | RMR | <0.05 | 0.000 | Pass |
| | RMSEA | <0.05 | 0.000 | Pass |
| | GFI | >0.9 | 1.000 | Pass |
| | AGFI | >0.9 | 1.000 | Pass |
| Relative fit index | IFI | >0.9 | 1.000 | Pass |
| | CFI | >0.9 | 1.000 | Pass |
| | NFI | >0.9 | 1.000 | Pass |
| | TLI | >0.9 | 1.002 | Pass |

Analysis of Table 12 reveals that the CMIN/DF value for testing the overall fitness of the model is 0.008, which is smaller than 3; the *p*-value is 0.928, which is greater than 0.05; RMR and RMSEA are both less than 0.05; and GFI, AGFI, IFI, CFI, NFI, and TLI are

all greater than 0.9. These results indicate that all test statistics meet the required fitness standards, that is, the results of the model are a good fit to the data.

Arrows were used to represent the influencing routes of mediators and the standardized estimated coefficient was adopted to show the degree of influence of different variables on EG. The functional routes of TII on EG are plotted according to Figure 7, as illustrated in Figure 8.

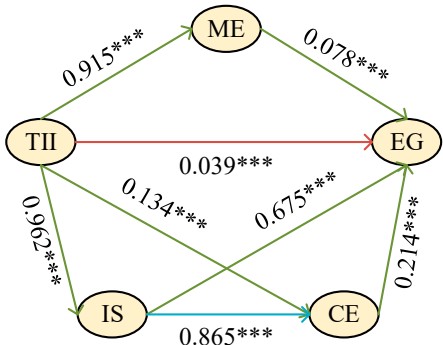

**Figure 8.** Functional routes of TII on EG. Notes: *** Significant at 1% level.

Analysis of Figure 8 shows that the route coefficients are all significant at the 1% statistical level, which means that hypotheses H1, H2, H2a, H2b, H3, H3a, H3b, H4, H4a, H4b, and H3-H4 are all accepted. To evaluate the relative importance of different independent variables to dependent variables, we calculate the standardized route coefficients by the method of maximum likelihood. Therein, the standardized route coefficients of TII → IS and TII → ME are, separately, 0.962 and 0.915, which indicate that TII mainly promotes EG indirectly through IS upgrading and stimulation of foreign trade. In addition, there is a significant route coefficient between TII and EG, which is consistent with the estimation results of the SDM, so the direct effect of TII on EG is further verified. Both the SDM and SEM models show the relationship between IS, CE and ME, and EG. It is reasonable to promote economic growth directly or indirectly through industrial structure, merchandize export, and consumer expenditure.

### 5.5.2. Bootstrap Tests on Function Routes

At first, the estimates of each function route were calculated according to the standardized estimated coefficient and, then, the rate of contribution of each complete function route was calculated based on the absolute value of the estimates. Then, 2000 rounds of repeated random sampling were simulated in Amos software and the bootstrap 95% confidence interval after error correction was outputted. The maximum likelihood estimation was adopted as the estimation method and bootstrap tests were conducted on the functional routes; results are listed in Table 13.

**Table 13.** Estimated results of functional routes of TII on EG.

| Functional Route | Estimate | Bootstrap Confidence Interval | $p$ Value | Rate of Contribution (%) |
|---|---|---|---|---|
| FR 1 | 0.040 | [0.019, 0.058] | 0.001 | 4.09 |
| FR 2 | 0.072 | [0.062, 0.084] | 0.001 | 7.37 |
| FR 3 | 0.657 | [0.615, 0.701] | 0.001 | 67.25 |
| FR 4 | 0.029 | [0.020, 0.040] | 0.001 | 2.97 |
| FR 5 | 0.180 | [0.145, 0.209] | 0.001 | 18.42 |
| Indirect effect | 0.937 | [0.906, 0.966] | 0.001 | 95.91 |
| Total effect | 0.977 | [0.948, 1.005] | 0.001 | 100.00 |

According to Table 13, the bootstrap 95% confidence interval of the direct and indirect effects does not include 0 and $p < 0.001$. ME, IS, and CE have a significant mediating effect,

while IS and CE have significant chained mediating effects. The result indicates that TII not only directly affects EG, but also can act on EG indirectly through IS, CE, and ME, and the indirect effects are greater than the direct effect. Suggesting that, when increasing direct investment, it is also necessary to pay close attention to the mediating effects of other modes on EG.

*5.6. Step 6: Analysis of the Functional Route Model*

To further explore the influencing mechanism of functional routes of TII on EG, functional routes FR1 to FR5 were expounded in detail based on bootstrap tests on the function route model.

FR1: TII → EG, in which the direct effect contributes 4.09%.

TII, as an investment element, can directly affect EG and amplifies its effect on EG through the multiplier effect. To be specific, improving the existing transport infrastructure and increasing the sharing and efficiency of transport infrastructure can expand demand and directly drive supply to the economy, thus, promoting EG.

FR2: TII → ME → EG, in which the indirect effect contributes 7.37%.

Enlarging the effective investment in transport infrastructure can maximize the positional advantages of transport infrastructure and narrow the gap in the development of transport infrastructure in various countries. This stimulates the potential impetus for foreign trade and becomes an important condition for expanding the export marginal effect and improving export comparative advantages of various countries. This also drives down logistics costs across the whole of society, unblocking the cycling of commodities, services, and production elements, and strengthening the vitality of economic development.

FR3: TII → IS → EG, in which the indirect effect contributes 67.25%.

The upgrading of IS is a key route for the effect of TII on EG. TII can exert an indirect effect on EG by influencing the upgrading of IS. Through the integration of elements, the favorable transport infrastructure lowers the trade cost and expands the market range of the service industry, thus, changing the IS. The adjustment of IS further changes resource allocation in the market and, thus, leads to EG.

FR4: TII → CE → EG, in which the indirect effect contributes 2.97%.

Thorough transport infrastructure provides convenience for residents' travel and affords people access to a larger scale of socioeconomic activities. Meanwhile, improving the quality of transport infrastructure can ensure travel safety and enhance personal willingness towards CE, changing the thinking behind consumption and consumer psychology, finally driving EG.

FR5: TII → IS → CE → EG, in which the indirect effect contributes 18.42%.

The high-quality development of the service industry promotes the upgrading of CE. The constant development of the service industry towards higher quality multiple categories provides diverse consumption choices to residents and enables the transformation of CE from traditional subsistence consumption to more spiritual consumption.

## 6. Conclusions

The theoretical framework for influences of transport infrastructure investment on economic growth was established in the research. The spatial Durbin model was used for empirical analysis of the spatial spillover effect of transport infrastructure investment on economic growth in 35 OECD+ countries from 2007 to 2019 to establish the function route model based on the structural equation model the further to explore the multiple mediating effects. The following conclusions can be drawn:

(1) The theoretical analysis mechanism was established to determine (in qualitative terms) the presence of the spatial spillover effect and multiple mediating effects of transport infrastructure investment in OECD+ countries on a theoretical basis. The comparative analysis model was built to quantify the difference in the effect of transport infrastructure investment on economic growth in OECD+ countries with different levels of income through empirical analysis. The effect of transport

infrastructure investment on economic growth is found to be greater in high-income countries than in upper-middle-income countries.

(2) According to empirical analysis of the spatial Durbin model, the spatial spillover effect of transport infrastructure investment differs significantly in OECD+ countries. In high-income OECD+ countries, transport infrastructure investment shows a positive spatial spillover effect, while the positive effect gradually diminishes with increasing geographical distance. In upper-middle-income OECD+ countries, transport infrastructure investment is found to have the negative spatial spillover effect. Although the increase in transport infrastructure investment is conducive to improving economic growth in a country, it intensifies the competition between upper-middle-income countries.

(3) According to empirical analysis of the function route model, transport infrastructure investment also exhibits multiple mediating effects in high-income OECD+ countries. Analysis of the total effect shows that transport infrastructure investment not only directly affects economic growth, but also indirectly influences economic growth through industrial structure and merchandize export and the indirect effects are greater than the direct effect. Therefore, the mediating effects of other modes also need to be paid attention to when increasing the direct investment. Analysis of indirect effects reveals that the FR3 has a rate of contribution of 67.25% (FR3 > FR5 > FR2 > FR4), so FR3 is identified as the key functional route, which means that transport infrastructure investment acts on economic growth mainly via indirect modes including industrial structure.

## 7. Policy Suggestions

The promotion of economic growth through transport infrastructure investment in OECD+ countries is a system-engineering problem. Based on the research conclusions, the following policy suggestions are proposed:

(1) The intensity of effective investment to transport infrastructure in OECD+ countries should be enhanced, and weak links in transport infrastructure construction and investment in burgeoning fields need to be focused on [1].When giving full play to the leading role, the government also needs to mobilize the vitality of social capitals and guide fund flows [73]. When timeously supplementing weak links in traditional transport infrastructure including the railway, highway, airport, harbor, and power sector systems, advanced technologies such as big data, the internet, and artificial intelligence should also be introduced [74]. This can improve the service efficiency of the comprehensive transport network and promote the upgrading of transport infrastructure using advanced information technology [75];

(2) Differentiated transport infrastructure investment strategies should be formulated according to the development of OECD+ countries with different levels of income [9]. High-income countries need to maintain the balanced development of transport infrastructure investment and make full use of the multilateral cooperation and development mechanism to foster economic ties between countries [11,76]. Upper-middle-income countries should start by improving regional transport infrastructure [20]. It is suggested that they establish a comprehensive transport network across sea, land, and air to provide a necessary spatial basis for facilitating export trade, to encourage flows of regional elements, and to improve regional competitiveness;

(3) The spatial spillover effect and multiple mediating effects of transport infrastructure investment on economic growth need to be given full play to improve the economic benefits of transport infrastructure investment by optimizing the layout and structure of transport infrastructure investment [15]. At present, transport infrastructure investment is in a stage of positive promotion in terms of its effect on economic growth in OECD+ countries and promoting structural adjustment of transport infrastructure investment can realize integration and connection of relevant industries of transport infrastructure [77]. The relevant department should pay particular attention to the pivotal role of industrial structure in the process in which trans-

port infrastructure investment drives economic growth and expands the range and service ability of services associated with transport infrastructure [78] to achieve coordinated development of transport infrastructure investment and industrial structure [40]. At the same time, functional routes of multiple factors including increasing the rate of growth of merchandize export, improving level of urbanization, and stimulating consumer expenditure should also be considered to achieve steady and sustainable economic development.

Economic growth and environmental sustainability are two of the most widely discussed topics today. While both of these issues are extremely important, it is widely believed that trying to solve one issue can impact the other. Though environmental sustainability is not addressed in this paper, it would be a good target for future research. In particular, we will actively explore ways to identify and mitigate multicollinearity and heteroskedasticity in future studies to improve the interpretability of data analysis.

**Author Contributions:** P.G. and J.F. proposed the original idea, as well P.G. and K.Z. organized the data collection. J.F. developed the theoretical part while P.G. and K.Z. developed the empirical model. K.Z. improved the English expression. All the authors provided critical feedback and helped shape the framework, analysis, and conclusion. All authors have read and agreed to the published version of the manuscript.

**Funding:** This study was supported by National Key R&D projects (Grant No: 2018YFC0704300), Hainan Major Science and Technology Plan Project (Grant No: ZDKJ202102021024), Major Science and Technology Projects of Sanya Yazhou Bay Science and Technology City (Grant No: SKJC-KJ-2019KY02), the Project of Sanya Yazhou Bay Science and Technology City (Grant No: SKJC-2022-PTDX-021), Hainan Special PhD Scientific Research Fund Project of Sanya Yazhou Bay Science and Technology City (Grant No: HSPHDSRF-2022-03-001), and China Scholarship Foundation (CSC No: 202206950011).

**Data Availability Statement:** Data of TII results from the transport infrastructure investment and maintenance spending of the International Transport Forum. Source: https://stats.oecd.org/Index.aspx?&datasetcode=ITF_INV-MTN_DATA, accessed on 1 January 2023. Data of GDP results from the World Development Indicators of the World Bank. Source: https://datatopics.worldbank.org/world-development-indicators/, accessed on 1 January 2023.

**Acknowledgments:** The authors of this study would like to thank STATA and AMOS software for their support in data computation.

**Conflicts of Interest:** The authors declare no conflict of interest.

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
