# Peer review of "The Spatial Spillover Effect and Function Routes of Transport Infrastructure Investment on Economic Growth: Evidence from Panel Data of OECD Members and Partners"

_mathematics, doi:10.3390/math11051167_

Round 1
Reviewer 1 Report
In my opinion, the reviewed article has a significant impact on the knowledge about the factors shaping economic growth and the role played by investments in transport infrastructure. In my opinion, the article is difficult to read due to the complexity of the issue and the tools used in the study. These tools, in particular SDM, SEM and partial differential equations (although the latter only verbally) are advanced enough to recommend an article for publication in Mathematics, despite the fact that it does not deal strictly with mathematics.
General thoughts:
The authors have a problem with the text editor page. There are double dots, sometimes spaces are missing. In the text, when referring to tables or figures, they sometimes use the word Table and a number, and sometimes also the title of the table. The latter approach is confusing and makes the article difficult to read. I encourage you to refer only to the name of the graphic element and its number. I encourage authors to review the entire article under the editor's account and correct these minor but annoying errors.
Titles of tables, graphs and figures also require improvement. They are mostly too laconic, short. The title should precisely define what is in the table. In many places there are no statements about what data is in the tables/figures, from what period of time, whether you use primary data or aggregates, e.g. averages. For example, Table 3 is titled "Descriptive Statistics". And what is it about? What variables? In what period? What is it for? I have referred to some of the titles in detailed comments. It is also worth describing the data sources.
I cannot speak authoritatively about the language used, but I have the impression that the text contains linguistic errors. It is worth that the authors should take care of improving the text also in this aspect before publishing it.
It is very tiring to use abbreviations in the text. SPD, SDM, SEM and many other models confuse the reader. I know that using full names would be pointless. Maybe it's worth writing down the full names at least in the key places of the text?
And about the SPD, SDM, SEM models. Maybe it's worth indicating somewhere in the text the sequence of creating these models and why some result from the other? This is probably described in the text, but the abbreviations got me so tired that I didn't notice it. I'm afraid other readers may have the same problem. Maybe a drawing, diagram or summary of modeling? Because in some models there are some variables, e.g. LI, UP and TR, in the next they disappear. And I don't know why this is happening.
Detailed notes:
line 40-42 is it OECD+? It's worth writing.
line 46 Table title. The table includes TII and GDP. In the title only TII. In addition, it is not known from which year these data are. It is necessary to precisely describe the content of this table in the title. The same goes for most of the next ones.
line 155 The program is called "Stata" not "Strata". A small but significant error. But there is a problem with this program in my opinion. You can't write that Stata calculated something, or you did the estimation using the Stata program. You got the right results thanks to mathematical and statistical tools. What you use for the calculations is secondary. After all, you will not write that you did some of the calculations using a Casio calculator or an abacus. They're just tools. It is important what procedures you use for calculations, inferences, etc. Once you can write somewhere that you use Stata and AMOS software to give credit to its authors and their copyrights, and that's enough.
line 243 Table 3 contains descriptive statistics (this title needs to be corrected). But why did you do it? In the text you do not refer to the data from this table. You don't describe her. Describe it somehow to justify its existence. Or, if you don't want to talk about the values of these stats, just drop the table from the text.
line 254 the title of figure 3 needs to be corrected. What year does this data refer to? What period actually. On the graph on the horizontal axes is EG. Why isn't there a word about it in the title. This chart does not compare TII across countries, but shows the relationship between TII and EG across countries by income. This needs to be described in the title.
Equation 1. Basically its description. The formula is marked W and later WD and WE are explained. The relationship between them is not explained. I can guess this, especially after studying the rest of the article. But you should write at least one task that WD and WE is a specification of the general notation of the W matrix for the two spaces you are interested in: geographical distance and economic distance (differences). If I understood it correctly. If not, then it needs to be explained even more.
line 288 what is the distance between countries? Distance between capitals, geographical centers, shortest distance between borders?
line 361 You wrote: the multiple mediation model is created by AMOS on the basis of SEM. It sounded like you had no idea how he did it. “You have a program, you push a button and something comes out. I don't know how it came about, but it's smart”. It was not AMOS that calculated (physically yes but it's just a tool) but some kind of research procedure. Talk about that, not the software.
Figure 5 the title needs to be written better. Now it doesn't say what the engraving is about. What's on this engraving? What data and in what period? TII and EG are average values?
line 415-416 you wrote that Moran's I for TII and EG show an increase over time. I don't see it on the figure. For TII there are some cyclical fluctuations and minimal growth. The graph for EG is basically constant. If you want to prove that it actually increased, maybe you need to use numbers? But why force the claim that there is an increase in the value of this index? Do you need it for something?
Figure 6 The title needs to be corrected. There are two years in the charts.
Table 6. Diagnostic tests. I believe these are the results of these diagnostic tests. What is SPD? Political party in Germany? In fact, it's probably explained on page 2. But what it is, it's not written there. And these shortcuts are tiring. You have to leaf through the entire article to explain the abbreviation. Maybe it is worth developing these abbreviations in such more important places, e.g. in the titles of chapters, tables and conclusions? Readers will be very grateful. And the description of stars in this table is missing. It's in the next one.
Table 7. I would add OECD+. For “OECD+ countries”. In the upper header, instead of H and UM, maybe it's worth expanding the abbreviations? There is room, they will fit. And it will be more readable.
Chapter 5.3.1. and 5.3.2. You have built models with many variables. In these models, many of the independent variables are statistically significant. But in the description you only talk about the significant influence of TII on EG. I understand that this is the content of the study, but you can't just focus on this relationship. There are variables in the model, but they are not in the description. Why? If you were only interested in the relationship between TII and EG, you had to make a single-variable model. You can't have many important variables but ignore them and only talk about one. They work together. It seems to me that the description of these two models should be improved by adding a description of the impact of the other variables. The more so that the SEM model confirms these dependencies.
line 455 where did the numbers 21.30% and 53.85% come from? They do not result directly from table 7, so you have to write how you calculated them.
Table 9. The title needs improvement. Details of the content of the table.
line 495 is it possible that the removal of one of the 13 measurements in the time series significantly changes the parameter estimates? Maybe. But your conclusion about the stability of the solution surprised me a bit. Why did you remove the first year? And not, for example, the last one. Have you checked what would happen to the model after removing the year 2019 instead of 2007? I'm not suggesting that 2019 would be better, but I'm asking why 2007. Why not two years?
line 526-528 Why is only CE and IS a mediator? Why didn't you use UP, LI and TR? They significantly affect the EG in the SDM model. It will probably be impossible to rebuild the route model, but at least explain why you rejected these three variables. Maybe I missed that in the text.
Table 11 and Figure 7 show basically the same thing. Is it worth doing this repetition? Do they bring something else? If you think so, then it might be. The engraving is nice. Though I don't understand column (5) in table 11.
line 563 is the word "supported" used correctly here?
line 563-566 What are normalized route factors? They have not been previously described. How should they be interpreted?
line 566-568 Both the SDM and SEM models show the relationship between IS, CE and ME and EG. Why don't you mention it? Their effect on EG seems to be stronger than TII's.
line 579-580 Maybe these conclusions should be supported with numbers?
number 581 Why not by CE? The results for CE are statistically significant.
line 605 Does the development of transport infrastructure affect IS? Or vice versa? You've written about this before, but I have serious doubts about this cause-effect relationship. As we build roads, services will develop. Or maybe when services develop, we will have to build roads? I have doubts about the order of events. The models only show the relationship, not its direction.
Expand the abbreviations in the conclusions. At least some. They don't look nice. Someone who only wants to read the introduction and conclusions will be bombarded with abbreviations and will not understand anything.
Author Response
We are very grateful to the Referees for their positive comments and their constructive points. Below is the point-by-point response with the changes we have implemented to address the specific points raised by Reviewer 1.
- line 40-42 is it OECD+? It's worth writing.
As suggested by the reviewer, we added a brief description on page 1 in the revised version: “Development in OECD+ countries as listed in Table 1, the gross domestic product (GDP) was 73.47 trillion dollars in 2019, accounting for 86.77% of global GDP, of which TII was 1.20 trillion.”
- line 46 Table title. The table includes TII and GDP. In the title only TII. In addition, it is not known from which year these data are. It is necessary to precisely describe the content of this table in the title. The same goes for most of the next ones.
As suggested by the reviewer, we changed the Table 1 title to “TII and GDP of OECD+ countries in 2019” and are glad to follow other suggestions.
- line 155 The program is called "Stata" not "Strata". A small but significant error. But there is a problem with this program in my opinion. You can't write that Stata calculated something, or you did the estimation using the Stata program. You got the right results thanks to mathematical and statistical tools. What you use for the calculations is secondary. After all, you will not write that you did some of the calculations using a Casio calculator or an abacus. They're just tools. It is important what procedures you use for calculations, inferences, etc. Once you can write somewhere that you use Stata and AMOS software to give credit to its authors and their copyrights, and that's enough.
As suggested by the reviewer, we have decided to delete the introduction of software tools, and added “The authors of this study would like to thank STATA and AMOS software for their support in data computation.” in the Acknowledgments part in the revised manuscript.
- line 243 Table 3 contains descriptive statistics (this title needs to be corrected). But why did you do it? In the text you do not refer to the data from this table. You don't describe her. Describe it somehow to justify its existence. Or, if you don't want to talk about the values of these stats, just drop the table from the text.
As suggested by the reviewer, we changed the Table 3 title to “Descriptive statistics of variables from 2007 to 2019” and we included an explanatory sentence in 3.2.1 in the revised version: “The standard deviation of EG is close to 1, indicating that the economic growth of different countries is significantly different. The standard deviation of TII, CE and ME is close to 2, shows that the development of merchandize export and consumer expenditure is not balanced in these countries. Comparatively speaking, there is no obvious difference between the development status of UP, IS, LI and TR in these countries.”
- line 254 the title of figure 3 needs to be corrected. What year does this data refer to? What period actually. On the graph on the horizontal axes is EG. Why isn't there a word about it in the title. This chart does not compare TII across countries, but shows the relationship between TII and EG across countries by income. This needs to be described in the title.
As suggested by the reviewer, we changed the Figure 3 title to “TII and EG in OECD+ countries with different income levels from 2007 to 2019” and are glad to follow other suggestions.
- Equation 1. Basically its description. The formula is marked W and later WD and WE are explained. The relationship between them is not explained. I can guess this, especially after studying the rest of the article. But you should write at least one task that WD and WE is a specification of the general notation of the W matrix for the two spaces you are interested in: geographical distance and economic distance (differences). If I understood it correctly. If not, then it needs to be explained even more.
As suggested by the reviewer, we explained the relation between W and WD 、WE on page 10 in the revised version: “Specifically, W can be estimated by geographical distance and economic distance.”
- line 288 what is the distance between countries? Distance between capitals, geographical centers, shortest distance between borders?
As suggested by the reviewer, we illustrated " dij denotes the regional distance calculated based on the longitude and latitude of the capital."
- line 361 You wrote: the multiple mediation model is created by AMOS on the basis of SEM. It sounded like you had no idea how he did it. “You have a program, you push a button and something comes out. I don't know how it came about, but it's smart”. It was not AMOS that calculated (physically yes but it's just a tool) but some kind of research procedure. Talk about that, not the software.
As suggested by the reviewer, we deleted the introduction of software tools.
- Figure 5 the title needs to be written better. Now it doesn't say what the engraving is about. What's on this engraving? What data and in what period? TII and EG are average values?
As suggested by the reviewer, we changed the Figure 5 title to “Aggregate trends and Moran’s I of TII and EG from 2007 to 2019” and are glad to follow other suggestions.
- line 415-416 you wrote that Moran's I for TII and EG show an increase over time. I don't see it on the figure. For TII there are some cyclical fluctuations and minimal growth. The graph for EG is basically constant. If you want to prove that it actually increased, maybe you need to use numbers? But why force the claim that there is an increase in the value of this index? Do you need it for something?
As suggested by the reviewer, we made a definite statement: “Moran’s I of TII increases in fluctuation, suggesting that the degree of spatial agglomeration of variables is enhanced.”
- Figure 6 The title needs to be corrected. There are two years in the charts.
As suggested by the reviewer, we changed the Figure 6 title to “Local Moran's I scatter plot of EG and TII in 2007 and 2019” and are glad to follow other suggestions.
- Table 6. Diagnostic tests. I believe these are the results of these diagnostic tests. What is SPD? Political party in Germany? In fact, it's probably explained on page 2. But what it is, it's not written there. And these shortcuts are tiring. You have to leaf through the entire article to explain the abbreviation. Maybe it is worth developing these abbreviations in such more important places, e.g. in the titles of chapters, tables and conclusions? Readers will be very grateful. And the description of stars in this table is missing. It's in the next one.
As suggested by the reviewer, we modified these shortcuts. And we have added the description of stars to table 6: “Notes: *** Significant at 1% level; ** significant at 5% level; * significant at 10% level”.
- Table 7. I would add OECD+. For “OECD+ countries”. In the upper header, instead of H and UM, maybe it's worth expanding the abbreviations? There is room, they will fit. And it will be more readable.
As suggested by the reviewer, we changed the Table 7 title to “Regression results of the SDM for OECD+ countries with different income levels” and modified these abbreviations in Table 7、Table 8、Table 9.
- Chapter 5.3.1. and 5.3.2. You have built models with many variables. In these models, many of the independent variables are statistically significant. But in the description you only talk about the significant influence of TII on EG. I understand that this is the content of the study, but you can't just focus on this relationship. There are variables in the model, but they are not in the description. Why? If you were only interested in the relationship between TII and EG, you had to make a single-variable model. You can't have many important variables but ignore them and only talk about one. They work together. It seems to me that the description of these two models should be improved by adding a description of the impact of the other variables. The more so that the SEM model confirms these dependencies.
As suggested by the reviewer, we added a brief description in chapter 5.3.1: “Besides, the level of urbanization and industrial structure both significantly stimulate EG and have a positive spatial spillover effect.”, and in chapter 5.3.2: “Besides, the total response of urbanization, industrial structure, labor input and trade both significantly increases in high-income countries under WE.”
- line 455 where did the numbers 21.30% and 53.85% come from? They do not result directly from table 7, so you have to write how you calculated them.
As suggested by the reviewer, we added the calculation processes: “Analysis of Table 7Table 7 implies that TII in high-income countries is underestimated by 21.30%: (0.108-0.085)/0.108=21.30%, while that in upper-middle-income countries is underestimated by 53.85%: (0.104-0.048)/0.104=53.85%, when only considering the geographical distance compared with coefficient estimation of WE.”
- Table 9. The title needs improvement. Details of the content of the table.
As suggested by the reviewer, we changed the Table 9 title to “The estimated results of robustness test based on shortening the time window” and are glad to follow other suggestions.
- line 495 is it possible that the removal of one of the 13 measurements in the time series significantly changes the parameter estimates? Maybe. But your conclusion about the stability of the solution surprised me a bit. Why did you remove the first year? And not, for example, the last one. Have you checked what would happen to the model after removing the year 2019 instead of 2007? I'm not suggesting that 2019 would be better, but I'm asking why 2007. Why not two years?
There are many ways to test robustness. We mainly consider that the statistical system and the work of data in the beginning year are not perfect, so we delete 2007 to test whether there are differences in the output results of the model.
- line 526-528 Why is only CE and IS a mediator? Why didn't you use UP, LI and TR? They significantly affect the EG in the SDM model. It will probably be impossible to rebuild the route model, but at least explain why you rejected these three variables. Maybe I missed that in the text.
As suggested by the reviewer, we added a brief description in chapter 5.4: “As shown in Table 3, the development of ME and CE is not balanced in these countries. And in Table 7, The value of IS is the largest and much larger than 1.”
- Table 11 and Figure 7 show basically the same thing. Is it worth doing this repetition? Do they bring something else? If you think so, then it might be. The engraving is nice. Though I don't understand column (5) in table 11.
Figure 7 shows the path structure and assumptions of the mediation effect model. Table 11 defines the direct effect, indirect effect and total effect, which we believe are conducive to readers quick understanding of the model construction. Both Figure 7 and Table 11 are necessary. The fifth column in Table 11 introduces the calculation method of the total effect, which is the preparation work for the calculation of path contribution rate in Table 13.
- line 563 is the word "supported" used correctly here?
As suggested by the reviewer, we modified "supported" to “accepted”.
- line 563-566 What are normalized route factors? They have not been previously described. How should they be interpreted?
As suggested by the reviewer, we added the definition of normalized route factors: “To evaluate the relative importance of different independent variables to dependent variables, we calculate the standardized route coefficients by the method of maximum likelihood.”
- line 566-568 Both the SDM and SEM models show the relationship between IS, CE and ME and EG. Why don't you mention it? Their effect on EG seems to be stronger than TII's.
As suggested by the reviewer, we added a brief description: “Both the SDM and SEM models show the relationship between IS, CE, ME and EG. It is reasonable to promote economic growth directly or indirectly through industrial structure, merchandize export and consumer expenditure.”
- line 579-580 Maybe these conclusions should be supported with numbers?
The significance of mediating effect is determined by P value: “According to Table 13, the bootstrap 95% confidence interval of the direct and indirect effects does not include 0 and P < 0.001. So ME, IS, CE have significant mediating effect, while IS and CE have significant chained mediating effects.”
- number 581 Why not by CE? The results for CE are statistically significant.
As suggested by the reviewer, we modified these description: “The result indicates that TII not only directly affect EG, but also can act on EG indirectly through IS, CE and ME, and the indirect effects are greater than the direct effect. Suggesting that, when increasing direct investment, it is also necessary to pay close attention to the mediating effects of other modes on EG.”
- line 605 Does the development of transport infrastructure affect IS? Or vice versa? You've written about this before, but I have serious doubts about this cause-effect relationship. As we build roads, services will develop. Or maybe when services develop, we will have to build roads? I have doubts about the order of events. The models only show the relationship, not its direction.
In FR3, we mainly analyzed that transport infrastructure investment promotes economic growth by influencing industrial structure, and does not conclude that industrial structure reacts on transport infrastructure. Here, we only analyze the path model of SEM.
- Expand the abbreviations in the conclusions. At least some. They don't look nice. Someone who only wants to read the introduction and conclusions will be bombarded with abbreviations and will not understand anything
As suggested by the reviewer, we expanded the abbreviations in the Conclusion and Policy suggestions.
The above contents are all our replies to the comments of reviewer 1. Please review them.
February 17, 2023

Reviewer 2 Report
Please find attached a file with my comments on the manuscript

Author Response
We are very grateful to the Referees for their positive comments and their constructive points. Below is the point-by-point response with the changes we have implemented to address the specific points raised by Reviewer 2.
- In the Abstract of the manuscript the authors refer to the "gross EG” (gross economic growth). I do not fully understand this term, given that economic growth refers by definition to the rate of change of Gros Domestic Product (GDP).
We were gratitude to Reviewer #2 for invaluable advice. Economic growth means an increase in production, production capacity, and all other components of an economy. The most common way to express economic growth is the rate of change of Gros Domestic Product (GDP) or a more actual GDP per capita. The annual average rate of change of the gross domestic product (GDP) at market prices based on constant local currency, for a given national economy, during a specified period of time. It expresses the difference between GDP values from one period to the next as a proportion of the GDP from the earlier period. GDP per capita is the most critical indicator of economic growth, shows the overall value production of all final goods and services in one country in a year. It follows that GDP per capita shows the ability of a society to meet its needs for the goods that are on the market as the subject of market exchange. Meanwhile, the choice of economic growth variable was based on these literatures.
[1] Jiang, X., et al., Multimodal transportation infrastructure investment and regional economic development: a structural equation modeling empirical analysis in China from 1986 to 2011. Transport Policy, 2017. 54: 43-52.
[2] Co A., Shs B., Mf A., Spatial relationship between economic growth and renewable energy consumption in 26 European countries. Energy Economics, 2020, 92: 104962.
[3] Junaid A., Liangqing L., Muhammad A.K., The Spillover Effects of Institutional Quality and Economic Openness on Economic Growth for the Belt and Road Initiative (BRI) countries, Spatial Statistics, 2022, 47: 100566.
- In Table 1, the authors present a list the countries examined. They include the high-income OECD countries and some upper-middle income OECD countries. Moreover, they include some other countries which they call “Key partners”. Finally, they include some countries, mainly upper-middle income, which they call “Observation countries”. With what criteria was their selection made? raise this point since there are many other countries, I consider significant partners in international trade that are not included in the table, such as Malaysia, Thailand, Iraq, Jordan and Libya.
“OECD Members, Key partners, Observation countries” were selected based on International Transport Forum (IMF) and World Bank. The investment situation of transport infrastructure in “Malaysia, Thailand, Iraq, Jordan and Libya” has not been counted by IMF, it is difficult to obtain data. Therefore, it is not calculated in this study, and we hope that a more perfect database can be used in future research.
- The analysis covers the period 2007-2019. It does not therefore include the period before the global financial crisis as well as the post-coronavirus period. We expect that the relationships between infrastructure investment and GDP have been affected especially in the post - covid period.
As suggested by the reviewer, the relationships between infrastructure investment and GDP have been affected especially in the post-covid period. But the statistical time range of IMF database is from 2000 to 2020, data from 2000 to 2006 and 2020 are seriously missing. Considering that SDM model is applicable to balanced panel data, we choose 2007 to 2019 as the research time. As the statistics are updated, we will fully examine the impact of the COVID in our next studies.
- The authors conclude that the spatial spillover effect in high income countries is positive while in upper-middle income is negative. If we consider the OECD members included in the analysis only three of them are upper-middle income (Colombia Mexico and Turkiye). The rest upper-middle income considered could be grouped as “economies in transition” (such as Bulgaria, Albania, Azerbaijan, Belarus, Georgia, Montenegro and Russian Federation) that have specific socioeconomic characteristics. These characteristics might affect the spatial spillover effect.
As suggested by the reviewer, we modified the original text and added sentences in 5.3.4 to reflect these situation: “It is worth noting that, Bulgaria, Albania, Azerbaijan, Belarus, Georgia, Montenegro and Russian Federation as OECD+ "economies in transition", due to its specific socioeconomic characteristics and the closeness of its connection with OECD countries, the spatial spillover effect of transport infrastructure investment is affected to different degrees.”
- The dependent variable in the model is GDP per capita (constant 2015 USS). I would rather consider the GDP per capita, PPP (constant 2017 international S), an indicator estimated by the World Bank, in its World Development Indicators database. Generally, real GDP per capita is not used in international comparisons, but rather the GDP per capita in PPP (purchasing power parities).
There are two calculation methods of GDP, namely international exchange rate and purchasing power parity. All calculation methods have limitations. On the one hand, international exchange rate cannot reflect the real purchasing power level; On the other hand, the main shortcoming of the theory of purchasing power parity is that it assumes that commodities can be traded freely. Besides, transaction costs such as tariffs, quotas and taxes are excluded, and the theory of purchasing power parity applies only to commodities, but ignores services, which may have a very significant value gap. Considering that the main measure of this study is the economic spillover effect of transportation infrastructure investment, the impact on services cannot be ignored, and it is also the focus of this study. Therefore, GDP per capita (constant 2015 USS) was chosen to measure economic growth.
- The initial conceptual model (Eq 1) starts as: In EGit = pW ln EGit + ... would rather consider that: ln EGit = pW ln EGit-1 + .. In other words, would examine the case of lagged dependent variable, since GDP per capita of one year might be affected by its value in the previous year.
Variable |
High-income |
Upper-middle-income |
||
|
|
|
|
|
ln TII |
0.108*** 1 |
0.085*** |
0.104*** |
0.048*** |
ln UP |
0.522*** |
0.571*** |
1.267*** |
1.072*** |
ln IS |
1.223*** |
2.383*** |
0.168** |
0.565*** |
ln LI |
0.713*** |
1.270*** |
-0.008 |
0.082* |
ln TR |
0.367*** |
0.291*** |
-0.029 |
-0.237*** |
L.W ln EG |
0.578*** |
-6.447*** |
-0.422*** |
-1.663*** |
W*ln TII 2 |
0.033 |
1.380*** |
-0.004 |
0.068*** |
W*ln UP |
-0.485** |
3.328*** |
0.236 |
1.917*** |
W* ln IS |
-0.687* |
24.878*** |
0.278*** |
1.178*** |
W*ln LI |
0.427*** |
7.039*** |
0.040 |
0.377*** |
W*ln TR |
0.037 |
3.615*** |
0.070 |
-0.450*** |
Spatial rho 3 |
0.033 |
0.076 |
0.013 |
0.038 |
Variance sigma2_e |
0.086*** |
0.092*** |
0.002*** |
0.001*** |
N |
324 |
324 |
96 |
96 |
r2 |
0.789 |
0.704 |
0.963 |
0.884 |
We fully agree with the views of the reviewer and thanks for your careful suggestions. In the empirical analysis, we considered the impact of economic development lag on the spatial spillover effect, but it was not reflected in Eq. (1), Eq. (8) and Eq. (9). We revised it. Thanks again for the suggestions of the reviewer.
- The authors state that “the logarithm of the GDP per capita on the abscissa represents the economic development level ...” The discussion regarding the extent to which GDP per capita is a satisfactory measure of economic development is long. The prevailing view is that GDP per capita is not a satisfactory measure of economic development. A number of indicators has been proposed. The most satisfactory one is the “human development index -HDI” proposed by the United Nations Development Programme – UNDP
We are glad to your suggestions. You have given me a lot of inspiration. Economic development is measured by multiple indicators and multiple dimensions. SDM model is characterized by only one dependent variable, so this study chooses GDP as the index of economic growth on the basis of summarizing the existing literature. According to your suggestions, the next research can adopt a comprehensive measurement approach and take "human development Index-HDI" as the measurement index system to measure economic growth in a more comprehensive way.
The above contents are all our replies to the comments of reviewer 2. Please review them.
February 17, 2023

Reviewer 3 Report
The authors have chosen a current topic that corresponds to the objectives of the journal, which can be discussed within the scope of an article. The authors proved that they are able to organize their study logically and organize it around a well-structured train of thought. Also, they are able to express themselves grammatically correctly and fluently, they use the established terms of their field accurately and consciously. The text of the article meets the stylistic requirements for the prose of a scientific lecture. The authors' style is sufficiently professional, but at the same time understandable.
The appearance of the authors' work is organized and clear, but at the same time, there are a significant number of figures and tables. There are no confusing typos or other errors in the text of the study. The study testifies that the authors tried to fully process the knowledge and documents related to their chosen topic. The literature used for the topic of the article is voluminous (it contains 82 items, topic specific. The list of literature used is synchronized with the references in the text, the same number of literature is included in the list of literature as is indicated in the text.
The strength of the study is that the model presented by the authors enables the formalization and identification of factors affecting economic growth in transport infrastructure developments.
The weakness of the article, on the other hand, is that it does not address at all the effects of the presented factors on the sustainability of transport, through which, in addition to the growth of the economy, the emission of greenhouse gases and the reduction of air pollution could become more effective in the future than at present. Another weakness of the article is that it does not present the problem of multicollinearity in linear regression models. In empirical analyses, however, it is a common case that not all data carry useful content from the point of view of the investigation, i.e. the data set is redundant. This phenomenon can be explained by multicollinearity. There are many ways to recognize and measure multicollinearity and to reduce the harmful consequences of this phenomenon. In addition, the econometric literature suggests additional estimation functions to eliminate heteroskedasticity. Among these, WLS stands out, which is based on the principle to minimize the sums of squares of differences weighted inversely with the variances. It would be worthwhile to apply this in the authors' model as well.
The article as a whole contains, in addition to the large amount of processed information, although not completely new findings, they undoubtedly contain interesting and significant results for scientific theory. At the same time, the study in its current form is too sophisticated for practice, it includes many concepts and methods that are little known to decision makers and difficult or impossible to understand.
Author Response
We are very grateful to the Referees for their positive comments and their constructive points. Below is the point-by-point response with the changes we have implemented to address the specific points raised by Reviewers 3.
- The weakness of the article, on the other hand, is that it does not address at all the effects of the presented factors on the sustainability of transport, through which, in addition to the growth of the economy, the emission of greenhouse gases and the reduction of air pollution could become more effective in the future than at present.
The reviewer's advice is forward-looking. Environmental pollution leads a rise in sustainable development problems. Greenhouse gases are one of the most important barriers against to sustainable development and greener cities. When the causes of greenhouse gases are investigated, human activities appeared as one of the main reasons. As one of the human activities, transportation has the highest impact on the increase in greenhouse gases emissions in green cities, such as air, railways, and road. Therefore, to provide a shed light for policymakers in implementing significant policies with specific targets to achieve low carbon emission target in this sector and other economic sectors, it is necessary to analyze the trend and characteristics of transport energy consumption and carbon emission policies and to decompose the main drivers of CO2 emission in this sector. This is also the focus of our team's research in the future.
- Another weakness of the article is that it does not present the problem of multicollinearity in linear regression models. In empirical analyses, however, it is a common case that not all data carry useful content from the point of view of the investigation, i.e. the data set is redundant. This phenomenon can be explained by multicollinearity. There are many ways to recognize and measure multicollinearity and to reduce the harmful consequences of this phenomenon. In addition, the econometric literature suggests additional estimation functions to eliminate heteroskedasticity. Among these, WLS stands out, which is based on the principle to minimize the sums of squares of differences weighted inversely with the variances. It would be worthwhile to apply this in the authors' model as well.
Just as the opinions of reviewers, multicollinearity and heteroscedasticity are common phenomena in regression models. The WLS method you proposed is a common means of data processing, and it is also one of the main methods we initially considered. In order to simplify the calculation process, we finally chose logarithmic processing. This is explained in chapter 6, pages 176-177, from Jeffrey Wooldridge's 《Introductory Econometrics A Modern Approach Fifth Edition》: Taking logarithms means the elasticity of the original explained variable with respect to the explained variable, the change in percentage rather than the change in value. Logarithmic form can reduce multicollinearity and eliminate the influence of dimension to a certain extent. In addition, logarithmic form can eliminate the influence of heteroscedasticity. Considering the large differences in sample data structure and in order to simplify the data processing process, logarithmic processing was carried out on sample data. Statistical indicators in Table 3 proved that sample data could be used for SDM modeling.
The above contents are all our replies to the comments of reviewer 3. Please review them.
February 17, 2023

Round 2
Reviewer 1 Report
I am satisfied with the answers and corrections. I have no further comments on the article. I recommend publishing the article in the presented form.
Congratulations to the authors of an interesting article. I wish you success in your further research work.
Author Response
Thank you very much for the detailed review and professional opinions of the reviewers, which provided a good opportunity for the revision and improvement of the paper.
We will continue to make efforts in future research and make more contributions to the development of scientific research in this field.

Reviewer 2 Report
The authors took into account all my points and provided satisfactory answers. Therefore, I propose the acceptance of the manuscript in its current form.
Sincerely yours,
Ioannis Vavouras
Author Response

(The authors gave the same response as above.)
